# Safely Learning Controlled Stochastic Dynamics

**Luc Brogat-Motte**
Laboratoire des Signaux et Systèmes, CNRS, CentraleSupélec
Université Paris-Saclay, Gif-sur-Yvette, France
Istituto Italiano di Tecnologia, Genoa, Italy
`luc.brogatmotte@iit.it`

**Alessandro Rudi**
SDA Bocconi, Bocconi University, Milan, Italy
`alessandro.rudi@sdabocconi.it`

**Riccardo Bonalli**
Laboratoire des Signaux et Systèmes, CNRS, CentraleSupélec
Université Paris-Saclay, Gif-sur-Yvette, France
`riccardo.bonalli@cnrs.fr`

## Abstract

We address the problem of safely learning controlled stochastic dynamics from discrete-time trajectory observations, ensuring system trajectories remain within predefined safe regions during both training and deployment. Safety-critical constraints of this kind are crucial in applications such as autonomous robotics, finance, and biomedicine. We introduce a method that ensures safe exploration and efficient estimation of system dynamics by iteratively expanding an initial known safe control set using kernel-based confidence bounds. After training, the learned model enables predictions of the system's dynamics and permits safety verification of any given control. Our approach requires only mild smoothness assumptions and access to an initial safe control set, enabling broad applicability to complex real-world systems. We provide theoretical guarantees for safety and derive adaptive learning rates that improve with increasing Sobolev regularity of the true dynamics. Experimental evaluations demonstrate the practical effectiveness of our method in terms of safety, estimation accuracy, and computational efficiency.

## 1 Introduction

We consider the problem of safely learning the dynamics of controlled continuous-time stochastic systems from discrete-time observations of trajectory data. This setting is common in applications such as robotics, finance, and healthcare, where system dynamics are only partially known and must be estimated from data. A key challenge in these applications is ensuring safety during both the learning phase and subsequent deployment [1; 2]. As an example, consider an autonomous robot navigating a partially known and turbulent environment, as illustrated in Figure 1. While the deterministic part of the dynamics may be approximately modeled using prior knowledge, the stochastic disturbances (represented by the brown region in Figure 1) due to wind or sensor noise are often unknown and must be learned. Collecting data through naive exploration can result in unsafe trajectories, potentially causing damage to the system or its environment. A second example arises in financial portfolio management, where the drift component of asset prices may be known from historical data, but market volatility remains uncertain. Safety here may correspond to the requirement

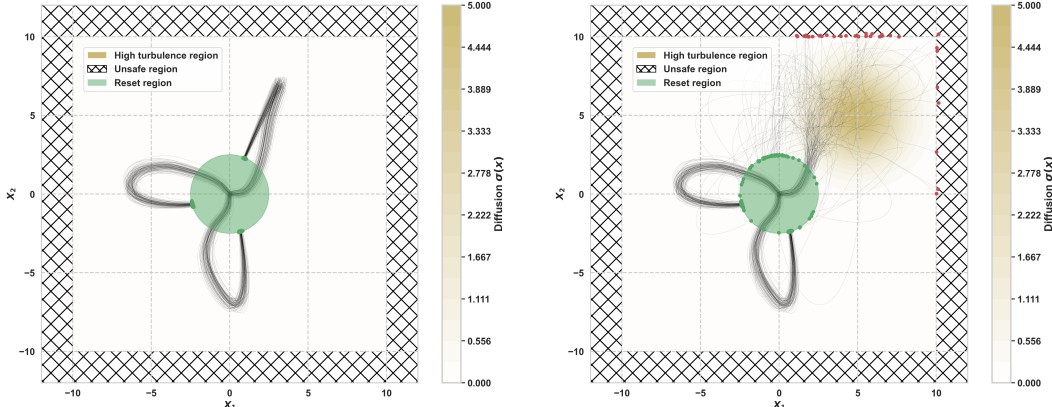

Figure 1: Illustration of a complex, smooth dynamical system under deterministic conditions (left) and stochastic conditions with unknown disturbances (right). Shown are 100 simulated trajectories under three different controls. Ignoring stochastic disturbances (e.g., wind turbulence) can lead to unsafe trajectories (right), emphasizing the necessity of safe estimation methods that explicitly account for uncertainty.

that the portfolio value stays above a critical threshold with high probability, simulating portfolio loss aversion in risk-sensitive financial decision-making. These examples highlight a common need: learning stochastic dynamics of a system from data, while ensuring safety throughout the process. This requires ensuring that all executed trajectories remain within a predefined safe region with high probability. In addition, at deployment time, the learned model should enable prediction of whether a proposed control input satisfies the safety requirements, including those not encountered during training [3; 4].

**Outline of contributions.** The contributions of this work are as follows.

- **Safe learning method.** We derive a method that safely learns controlled stochastic dynamics, where safety is defined as the requirement that system trajectories remain within a designated set of safe states with high probability. Our approach incrementally expands the known safe control set by selecting novel controls to evaluate, collecting corresponding trajectory data, and refining three models: a dynamics model for predicting state evolution, a safety model that estimates the probability of remaining within the safe region, and a reset model that captures the probability of returning the system to its initial state distribution, enabling repeated safe exploration under uncertainty. Alongside these models, we refine uncertainty estimates using kernel-based confidence bounds. After training, these models enable prediction of dynamics, safety, and reset feasibility for any given control, including those not seen during training.

- **Provably safe exploration and adaptive estimation rates.** We prove that the proposed method guarantees safety and derive learning rates for system estimation, with rates that are adaptive to the Sobolev regularity of the underlying dynamics. Crucially, our approach requires only smooth dynamics (with respect to time, state, and control variables) and an initial non-empty set of known safe controls. These mild assumptions make the method applicable to a broad range of complex real-world systems subject to stochastic disturbances, found in diverse areas including robotics, fluid flow control, and chemical reaction control [5; 6; 7; 8; 9; 10].

- **Experimental validation.** We empirically demonstrate the performance of the approach in terms of safety, estimation accuracy, and computational efficiency. Specifically, we evaluate it on a benchmark two-dimensional stochastic dynamical system evolving in a bounded, safety-critical environment under stochastic perturbations (see Figure 1). An open-source Python implementation is provided (available at `github.com/lmotte/dynamics-safe-learn`).

## 2 Background

We formalize the problem of safely learning controlled stochastic dynamical systems as follows.

**Controlled SDE.** Let $X$ be a dynamical system governed by a non-linear, $n$-dimensional controlled SDE [11; 12],

$$dX(t) = b(X(t), u(t, X(t)))dt + a(X(t), u(t, X(t)))dW(t), \quad u \in \mathcal{H}, \quad X(0) \sim p_0. \quad (1)$$

Here, $T_{\max} > 0$ is the fixed time horizon, $(b, a)$ are functions mapping $\mathbb{R}^n \times \mathbb{R}^d$ to $\mathbb{R}^n \times \mathbb{R}^{n \times n}$, $W(t)$ is an $n$-dimensional standard Brownian motion, $p_0$ is an initial probability density over $\mathbb{R}^n$, and $\mathcal{H} \subseteq \mathcal{F}([0, T_{\max}] \times \mathbb{R}^n, \mathbb{R}^d)$ is a finite-dimensional set of admissible controls, where $\mathcal{F}([0, T_{\max}] \times \mathbb{R}^n, \mathbb{R}^d)$ denotes the space of measurable functions from $[0, T_{\max}] \times \mathbb{R}^n$ to $\mathbb{R}^d$.

**Example 1** (Second-order dynamical system). *The system*

$$dX(t) = V(t) \, dt, \quad dV(t) = u(t, X(t)) \, dt + a(X(t)) \, dW(t),$$

*captures second-order dynamics under control $u$ and state-dependent diffusion $a(X(t))$. Real-world examples include drones navigating turbulent environments (where $u$ represents thrust or steering, and $a(\cdot)$ models wind turbulence), fluid-dynamical systems [13], and molecular dynamics (where $u$ captures external forces such as optical traps, and $a(\cdot)$ reflects spatially varying thermal fluctuations [14; 15; 16]).*

**Safety-critical environments.** Let $g : \mathbb{R}^n \to \mathbb{R}$ be a function that partitions the state space into safe and unsafe regions. The *safe region* is given by $\{x \in \mathbb{R}^n : g(x) \geq 0\}$, while the *unsafe region* is given by $\{x \in \mathbb{R}^n : g(x) < 0\}$.

**Safe control.** Let $X_u$ denote the solution to Eq. (1) under the control $u \in \mathcal{H}$ (well-defined by existence and uniqueness; see Sec.3 in [17]). Let $u_\theta : [0, T_{\max}] \times \mathbb{R}^n \to \mathbb{R}^d$ denote a family of controls parameterized by $\theta \in D \subset \mathbb{R}^m$, where $D$ is a compact subset of $\mathbb{R}^m$. Such a finite-dimensional parameterization is a mild assumption that is widely prevalent in many real-world applications, including robotics [18], process control [19], and financial engineering [20].

We define the safety level of the control $u_\theta$ *at* time $t \in [0, T_{\max}]$ as

$$s(\theta, t) \triangleq \mathbb{P}(g(X_{u_\theta}(t)) \geq 0). \quad (2)$$

We define the safety level *up to* time $T \in [0, T_{\max}]$ of the control $u_\theta$ as

$$s^\infty(\theta, T) \triangleq \inf_{t \in [0, T]} s(\theta, t). \quad (3)$$

**Learning problem.** We formulate the problem of safely learning controlled stochastic dynamical systems as the estimation of the probability density map

$$p : (\theta, t, x) \in D \times [0, T_{\max}] \times \mathbb{R}^n \mapsto p_\theta(t, x), \quad (4)$$

where $p_\theta(t, x)$ is the density of the state $X_{u_\theta}(t)$ under control $u_\theta$, well-defined by standard existence and uniqueness results (see Sec.3 in [17]). To estimate this density, we collect a dataset of trajectories

$$(\theta_k, X_{u_{\theta_k}}(w_i^k, t_l))_{k \in \{1, \dots, K\}, \, i \in \{1, \dots, Q\}, \, l \in \{1, \dots, M_k\}}, \quad (5)$$

where each $w_i^k$ denotes an independent Brownian motion sample path driving the stochastic trajectory, and $M_k$ denotes the number of time steps in trajectory $k$. All controls $u_{\theta_k}$ are required to be safe. Specifically, the minimal probability of remaining within the safe region up to the time horizons $(T_k)_{k=1}^K = (t_{M_k})_{k=1}^K$ is constrained by

$$s^\infty(\theta_k, T_k) \geq 1 - \varepsilon, \quad \text{for each } k \in [\![1, K]\!]. \quad (6)$$

This problem poses a fundamental challenge due to the coupling between learning and safety: accurately estimating the density $p_\theta$ requires data, but collecting data must respect safety constraints defined by $s^\infty$, which themselves depend on the very dynamics encoded in $p_\theta$.

## 2.1 Related work

Safe learning in control systems under uncertainty is a central topic in reinforcement learning, with methods built on assumptions such as known dynamics [21], controllability [22; 23; 24], or recovery policies [25; 26].

Much of the literature focuses on discrete-time or discrete-state systems modeled as Markov Decision Processes (MDPs), where risk-sensitive and safe exploration techniques have been developed [27; 28; 26; 29]. Among these, Safe Bayesian Optimization (BO) methods stand out for providing some of the strongest high-confidence safety guarantees during exploration [30; 31; 32; 33], particularly in MDP settings with known dynamics or access to safety level evaluations [22; 34]. In continuous domains, early work focused on designing control policies that avoid unsafe regions [35; 36], while more recent approaches incorporate offline learning and online adaptation for nonlinear systems [37]. Stability-based guarantees via Lyapunov theory offer formal certification but often require full dynamics knowledge [18; 38]. Safe BO has also been adapted to continuous settings [39; 40], under assumptions such as access to dynamics, safety oracles, or specific control-theoretic properties. Joint estimation of both dynamics and safety remains comparatively underexplored, particularly in continuous-time settings [41; 42; 43; 44; 45; 46; 47; 48; 49].

In contrast to much of the literature, we assume no prior model of the system dynamics or safety function. Instead, we jointly explore and learn both the stochastic dynamics and the safety probabilities from trajectory data. Our method applies to broad classes of continuous-time, continuous-state stochastic systems, and provides provable guarantees on both safety and estimation accuracy. To the best of our knowledge, no prior work provides joint safe exploration and density estimation guarantees in this setting.

## 3 Assumptions

As formalized by the No-Free-Lunch Theorem [50], learning is only possible under prior assumptions. We now state and discuss the key assumptions used in this work.

**Assumption (A1)** (Initial safe controls). *For $\varepsilon \in [0, 1]$, a non-empty set $S_0 \subset D \times [0, T_{\max}]$ is provided such that*

$$s(\theta, t) \geq 1 - \varepsilon \quad \text{for all} \quad (\theta, t) \in S_0. \tag{7}$$

This assumption ensures that at least one control is known to be safe at the outset, allowing safe exploration to begin. In fact, $S_0$ may be as small as a singleton; only one known safe control is required. Without such a point, safe learning cannot be initiated. We express the assumption in set form to allow for larger safe sets, which can accelerate exploration while preserving guarantees. This is a standard assumption in the literature of safe UCB methods [30; 22; 18], and is realistic in many applications including robotics [39] and safety-critical process control [51], where systems naturally start in safe conditions, e.g., $S_0 = \{(0, \theta) : \theta \in D\}$.

**Resetting control.** Let $h : \mathbb{R}^n \to \mathbb{R}$ define a region in the state space from which resets are feasible. Specifically, if $h(X(t)) \geq 0$, then it is feasible to reset the system to the initial distribution $p_0$. Formally, this means there exists a mechanism (or control) that reinitializes the system from the current state $X(t)$ to a new state independently sampled from $p_0$. We define the reset level at time $t \in [0, T_{\max}]$ for a given control $u_\theta$ as

$$r(\theta, t) \triangleq \mathbb{P}\left(h(X_{u_\theta}(t)) \geq 0\right). \tag{8}$$

The function $h$ delimits a region of the state space from which resets are feasible. Larger reset regions correspond to greater operational flexibility. In simulated environments, where resets are effectively cost-free, the reset region can cover the entire state space $\mathbb{R}^n$. In contrast, real-world systems typically require substantial resources or manual intervention, making resets feasible only in restricted regions (e.g., near the original distribution $p_0$).

**Assumption (A2)** (Initial resetting controls). *For $\xi \in [0, 1]$, a non-empty set $R_0 \subset D \times [0, T_{\max}]$ is provided such that*

$$r(\theta, t) \geq 1 - \xi \quad \text{for all} \quad (\theta, t) \in R_0. \tag{9}$$

This assumption guarantees the existence of at least one control capable of returning the system to the reset region with high probability. Assumption (A2) enables the generation of independent sample paths starting from the same initial conditions, which is crucial for evaluating variance and managing uncertainty during safe exploration. As with Assumption (A1), one known reset point is sufficient, though larger reset sets accelerate learning. A simple case is $R_0 = \{(0, \theta) : \theta \in D\}$, corresponding to systems that can always be reset from the initial condition. In practice, reset feasibility depends on system constraints: for instance, batch chemical reactors can often be reset only in early phases, before irreversible reactions occur [19]. In contrast, many autonomous systems like drones or driving robots can usually be reset, at least during training.

**Assumption (A3)** (Smoothness of system dynamics). *The map $p$ lies in the Sobolev space $H^\nu(\mathbb{R}^{n+m+1})$ with $\nu > \frac{1}{2}\max(n, m+1)$, where $n$ and $m$ denote the state and control parameter dimensions, respectively. Moreover, $\sup_{x \in \mathbb{R}^n} \left\| p(\cdot, \cdot, x) \right\|_{H^\nu(\mathbb{R}^{m+1})} < +\infty$, $\sup_{(\theta,t) \in D \times [0,T_{\max}]} \left\| p(\theta, t, \cdot) \right\|_{H^\nu(\mathbb{R}^n)} < \infty$.*

This smoothness assumption ensures that the system dynamics are sufficiently regular for our purposes. It is standard in statistical learning theory and underpins our convergence guarantees [52]. Sobolev regularity of the drift and diffusion terms in the underlying stochastic differential equation is expected to imply Sobolev regularity of the resulting state densities under standard conditions. This follows from classical results in parabolic PDE theory, where solutions typically gain regularity relative to the coefficients, roughly two derivatives in space and one in time. A formal analysis of this connection is beyond the scope of the present work and is left for future investigation (see Bonalli and Rudi [17] for related results).

# 4 Proposed method

We propose a method for safely exploring and learning system dynamics over a parameterized control space $\mathcal{H} = \{u_\theta \mid \theta \in D \subset \mathbb{R}^m\}$. Following the safe UCB framework [30; 31; 32], our goal is to select controls that reduce model uncertainty while ensuring, with high probability, that trajectories (i) remain within the safe region and (ii) end in the reset region. We jointly learn three models: a dynamics model (state densities), a safety model (safety probabilities), and a reset model (reset probabilities), each equipped with confidence bounds from a shared kernel. This enables active exploration under high-probability constraints. After training, the learned models support inference on unseen inputs and yield a certified control set that can be deployed with safety guarantees.

The known safe-resettable set is expanded iteratively by alternating between system estimation (Section 4.2) and safe sampling (Section 4.3), leveraging prior knowledge of the initial safe and reset sets as well as the regularity of the dynamics $(\theta, t, x) \mapsto p_\theta(t, x)$.

A step-by-step breakdown of the overall method is provided in Appendix B, with algorithm tables for each module and their computational complexities.

## 4.1 Initialization

Let $N \in \mathbb{N}$ denote the current iteration. We initialize at $N = 0$ using the known safe set $S_0 \subset D \times [0, T_{\max}]$ and reset set $R_0 \subset D \times [0, T_{\max}]$. We define the initial safe-resettable set

$$\Gamma_0 \triangleq \left\{ (\theta, t, T) \in D \times [0, T_{\max}]^2 \, \middle| \, t \leq T, \ (\theta, t') \in S_0 \text{ for all } t' \in [0, T], \ (\theta, T) \in R_0 \right\}.$$

We select $(\theta_0, t_0, T_0) \in \Gamma_0$, ensuring that the control is known to be safe over $[0, T_0]$ and ends in the reset region.

## 4.2 System estimation

In this step, we update the dynamics, safety, and reset models based on the observed trajectories, and compute predictive uncertainty for each.

**Estimation at $(\theta_N, t_N)$.** At iteration $N$, the control $u_{\theta_N}$ is evaluated using $Q$ stochastic trajectories. Here, $N$ indexes the iteration, and $i$ indexes the $i$-th trajectory simulated under that control, each corresponding to an independent sample $w_i^N$ of the Brownian motion driving the system. We collect

the samples $(X_{u_{\theta_N}}(w_i^N, t_N))_{i=1}^Q$ and estimate the state density at $(\theta_N, t_N)$ using a kernel density estimator:

$$\hat{p}_{\theta_N, t_N}(x) \triangleq \frac{1}{Q} \sum_{i=1}^Q \rho_R(x - X_{u_{\theta_N}}(w_i^N, t_N)), \tag{10}$$

where $\rho_R(x) \triangleq R^{n/2} \|x\|^{-n/2} B_{n/2}(2\pi R\|x\|)$, $R > 0$, and $B_{n/2}$ is the Bessel $J$ function of order $n/2$ (See Bonalli and Rudi [17]).

We then compute estimates of the safety and reset probabilities

$$\hat{s}_{\theta_N, t_N} \triangleq \int_{\{x \in \mathbb{R}^n : g(x) \geq 0\}} \hat{p}_{\theta_N}(t_N, x)\, dx, \quad \hat{r}_{\theta_N, t_N} \triangleq \int_{\{x \in \mathbb{R}^n : h(x) \geq 0\}} \hat{p}_{\theta_N}(t_N, x)\, dx. \tag{11}$$

Let the collection of values at all observed points $((\theta_i, t_i))_{i=1}^N$ be

$$\hat{P}(\cdot) \triangleq (\hat{p}_{\theta_i, t_i}(\cdot))_{i=1}^N, \quad \hat{S} \triangleq (\hat{s}_{\theta_i, t_i})_{i=1}^N, \quad \hat{R} \triangleq (\hat{r}_{\theta_i, t_i})_{i=1}^N.$$

**Model update.** We fit kernel ridge regressors for the density, safety, and reset functions using a Matérn kernel $k$ (with Sobolev smoothness $\nu$) and regularization $\lambda > 0$

$$\hat{p}_\theta(t, x) \triangleq \hat{P}(x)(K + N\lambda I)^{-1} k(\theta, t), \qquad \text{(System dynamics)} \tag{12}$$

$$\hat{s}_N(\theta, t) \triangleq \hat{S}(K + N\lambda I)^{-1} k(\theta, t), \qquad \text{(Safety function)} \tag{13}$$

$$\hat{r}_N(\theta, t) \triangleq \hat{R}(K + N\lambda I)^{-1} k(\theta, t), \qquad \text{(Reset function)} \tag{14}$$

where $k(\theta, t) \triangleq (k((\theta, t), (\theta_i, t_i)))_{i=1}^N$, $K \triangleq (k((\theta_i, t_i), (\theta_j, t_j)))_{i,j=1}^N$, and $\lambda$ is a regularization term. Although training data consist of discrete-time observations, learned regression models are defined over continuous time, a distinction seldom addressed in the literature.

The predictive uncertainty at $(\theta, t)$ is given by

$$\sigma_N^2(\theta, t) \triangleq k((\theta, t), (\theta, t)) - k(\theta, t)^*(K + N\lambda I)^{-1} k(\theta, t). \tag{15}$$

### 4.3 Safe sampling

We now select a new point to sample by maximizing uncertainty over the safe-resettable region.

**Feasibility criteria.** A point $(\theta, t, T)$ is feasible if (i) $t \leq T$, (ii) the system remains safe up to time $T$, i.e., $s^\infty(\theta, T) \geq 1 - \varepsilon$, and (iii) the trajectory ends in the reset region with high probability, i.e., $r(\theta, T) \geq 1 - \xi$.

We implement these constraints via lower confidence bounds (LCBs)

$$\mathrm{LCB}_N^s(\theta, T) \triangleq \inf_{t \in [0, T]} \left( \hat{s}_N(\theta, t) - \beta_N^s \sigma_N(\theta, t) \right), \tag{16}$$

$$\mathrm{LCB}_N^r(\theta, T) \triangleq \hat{r}_N(\theta, T) - \beta_N^r \sigma_N(\theta, T), \tag{17}$$

where $\beta_N^s, \beta_N^r > 0$ are confidence parameters set from known upper bounds on the RKHS norms of the safety and reset functions (see Remark 1).

We then define the safe-resettable feasible set

$$\Gamma_N = \Gamma_0 \cup \left\{ (\theta, t, T) \in D \times [0, T_{\max}]^2 \mid t \leq T,\ \mathrm{LCB}_N^s(\theta, T) \geq 1 - \varepsilon,\ \mathrm{LCB}_N^r(\theta, T) \geq 1 - \xi \right\}.$$

**Sampling rule.** We choose the next $(\theta_{N+1}, t_{N+1}, T_{N+1})$ by maximizing uncertainty over the feasible set

$$(\theta_{N+1}, t_{N+1}, T_{N+1}) = \underset{(\theta, t, T) \in \Gamma_N}{\arg\max} \ \sigma_N(\theta, t). \tag{18}$$

Optimization is performed using discretization or gradient-based methods. Several computational techniques for efficient optimization are presented in the Appendix (see Algorithm 4).

**Stopping rule.** We stop the exploration once the maximum uncertainty over the feasible set falls below a threshold $\eta > 0$

$$\max_{(\theta,t,T)\in\Gamma_N} \sigma_N(\theta,t) < \eta, \tag{19}$$

ensuring that exploration concludes once the models reaches the desired level of accuracy.

**Remark 1.** *The derivation in Appendix A.7 shows that the confidence parameters depend on upper bounds of the RKHS norms of the safety and reset functions. These bounds may be available from prior knowledge of the system's regularity. We view this prior knowledge as a reasonable minimal assumption for guaranteeing safe learning under unknown dynamics. When unavailable, the bounds can be conservatively overestimated, ensuring safety but potentially leading to slower exploration. Developing adaptive strategies to estimate these quantities without prior knowledge is a promising direction for future work, for instance through online adaptation via the doubling trick [53].*

## 5   Safety and estimation guarantees for Sobolev dynamics

Safety and exploration guarantees for safe kernelized UCB methods have been developed in prior work [30; 31; 32], grounded in kernelized bandit theory [54; 55; 56], which in turn builds on linear bandit results [57; 58]. Building on this foundation, we establish novel theoretical guarantees for safe exploration and dynamics estimation under Sobolev regularity. Complete proofs are deferred to Appendix A.

**Theorem 5.1** (Safely learning controlled Sobolev dynamics)**.** *Let $\eta > 0$, and assume Assumptions (A1)–(A3) hold. Set $R = Q^{1/(n+2\nu)}$ and $\lambda = N^{-1}$. Then there exist constants $c_1,\ldots,c_5 > 0$, independent of $N, Q, \delta, \eta$, such that if*

$$c_1 \log(4N/\delta)^{1/2} Q^{\frac{n-2\nu}{2n+4\nu}} \leq N^{-1/2},\text{[1]}$$

*then the stopping condition $\max_{(\theta,t,T)\in\Gamma_N} \sigma_N(\theta,t) < \eta$ is satisfied after at most $N \leq c_2\eta^{-2/(1-\alpha)}$ iterations for any $\alpha > (m+1)/(m+1+2\nu)$. Moreover:*

- *(Safety): All selected triples $(\theta_i, t_i, T_i)$ satisfy $s^\infty(\theta_i, T_i) \geq 1 - \varepsilon$ and $r(\theta_i, T_i) \geq 1 - \xi$, providing safety guarantees during training. Moreover, the final set $\Gamma_N$ includes only controls meeting these thresholds and can thus serve as a certified safe set for deployment.*

- *(Estimation guarantees): For all $(\theta, t, T) \in \Gamma_N$,*

$$\|\hat{p}_\theta(t,\cdot) - p_\theta(t,\cdot)\|_\infty \leq c_3\eta, \quad |\hat{s}_N(\theta,t) - s(\theta,t)| \leq c_4\eta, \quad |\hat{r}_N(\theta,t) - r(\theta,t)| \leq c_5\eta.$$

This result ensures that our method both respects safety constraints and achieves convergence rates adaptive to the system's Sobolev regularity. The condition on $Q$ provides a lower bound on the number of trajectory samples $Q$ required per control to guarantee the prescribed confidence level. Up to logarithmic factors, it requires

$$Q \gtrsim N^{\frac{2\nu+n}{2\nu-n}},$$

where $n$ is the state dimension and $\nu$ the Sobolev regularity. For instance, when the system is sufficiently regular with $\nu \geq n + m + 1$, the algorithm terminates in at most $N = \mathcal{O}(\eta^{-3})$ iterations, assuming $Q \gtrsim N^3$. Although we do not analyze the size of $\Gamma_N$, its structure can be inferred from the available uncertainty estimates; a formal characterization of $\Gamma_N$ is left to future work.

## 6   Numerical experiments

We evaluate our method on a representative smooth nonlinear stochastic system. Specifically, the experiments aim to assess the following: (i) satisfaction of safety and reset constraints, (ii) efficiency of exploration under different safety thresholds, (iii) prediction accuracy for dynamics, safety, and reset maps, (iv) computational cost and scalability.

---

[1]All exponents in $n$ are to be read as $n + \varepsilon$, with $\varepsilon > 0$ arbitrarily small.

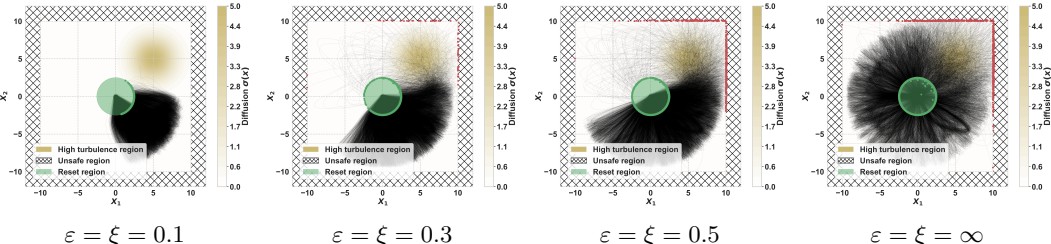

$$\varepsilon = \xi = 0.1 \qquad \varepsilon = \xi = 0.3 \qquad \varepsilon = \xi = 0.5 \qquad \varepsilon = \xi = \infty$$

Figure 2: One trajectory per selected control after 1000 iterations for various thresholds.

**System and environment.** We consider a 2D second-order dynamical system whose acceleration is directly controlled by the input. The system evolves according to the controlled SDE

$$\begin{cases} dX(t) = V(t)dt, \\ dV(t) = u(t, X(t), V(t))dt + a(X(t))dW_t. \end{cases}$$

where $X(t) \in \mathbb{R}^2$, $V(t) \in \mathbb{R}^2$ denote position and velocity, $u$ is the control function, and $W_t$ is a Brownian motion. The noise amplitude $a(X)$ is spatially dependent:

$$a(X) = A \exp\left( -\frac{\|X - X_c\|^2}{2\sigma^2} \right),$$

with $X_c = (5,5)$, $\sigma = 2$, and $A = 5$. The initial state follows $\mathcal{N}(0, \sigma_0 I_{\mathbb{R}^2})$ with $\sigma_0 = 0.1$, and the maximal time horizon is $T_{\max} = 20$. The system evolves within the bounded safe region $(-10, 10)^2$, and each trajectory must end in the reset region defined as a disk of radius 2.5 centered at the origin. Such models arise in robotics and autonomous navigation, involving trajectory control with localized disturbances (e.g., slippery or uneven terrain). Figure 1 illustrates the effect of such state-dependent noise through 100 trajectories generated under different controls.

**Control space.** Controls are parameterized as sequences of $m$ fixed accelerations of magnitude $v$, applied in directions $(\theta_1, \ldots, \theta_m)$. During the exploration phase $(0 \leq t \leq T_{\text{explo}})$, each direction $\theta_i$ is applied over intervals of equal length, yielding

$$u(t, X, V) = v(\cos(\theta_i), \sin(\theta_i)) - V,$$

with damping term $-V$ ensuring velocity convergence. For $(t > T_{\text{explo}})$, a feedback controller steers the system toward $\mu_0$:

$$u(t, X, V) = \kappa \times \left( v\frac{\mu_0 - X}{\|\mu_0 - X\|} - V \right),$$

with damping factor $\kappa > 0$. Controls are clipped to keep the system within the safe region. We set $v = 2.0$, $\kappa = 0.5$, $m = 2$, $T_{\text{explo}} = 6$, and $n_{\text{steps}} = 500$.

**Method's hyperparameters.** Our method depends on several hyperparameters that govern safety thresholds $(\varepsilon, \xi)$, confidence levels $(\beta_s, \beta_r)$, kernel smoothness $(\lambda, \gamma)$, and bandwidth $R$, with distinct values for estimating dynamics and constraints. We test $(\varepsilon, \xi) \in \{0.1, 0.3, 0.5, \infty\}$, with 1000 iterations and initial safe control $(-\pi/3, \pi/3)$. Candidate selection for uncertainty maximization is restricted to a local subset for efficiency (Appendix B, Algorithm 4). A detailed discussion of each hyperparameter's role, tuning procedure, and practical heuristics is provided in Appendix B.3.

**Safe exploration.** Figure 2 displays one trajectory per selected control after 1000 iterations, under various threshold settings ($\varepsilon = \xi \in \{0.1, 0.3, 0.5, +\infty\}$). In Figure 3, the top row displays the learned safety level maps while the bottom row shows the corresponding reset probability maps for various threshold pairs $\varepsilon = \xi$, with values increasing from left to right in $\{0.1, 0.3, 0.5, +\infty\}$. Our approach only accepts candidate controls whose predicted safety and reset probabilities (estimated via 200-path Monte Carlo simulations) exceed the predefined thresholds. By filtering only controls meeting the safety criteria, exploration is confined to a safe region with a chosen probability of staying safe. Overall, these visualizations highlight how increasing the threshold values influences control selection, providing insights into the trade-off between exploration and safety.

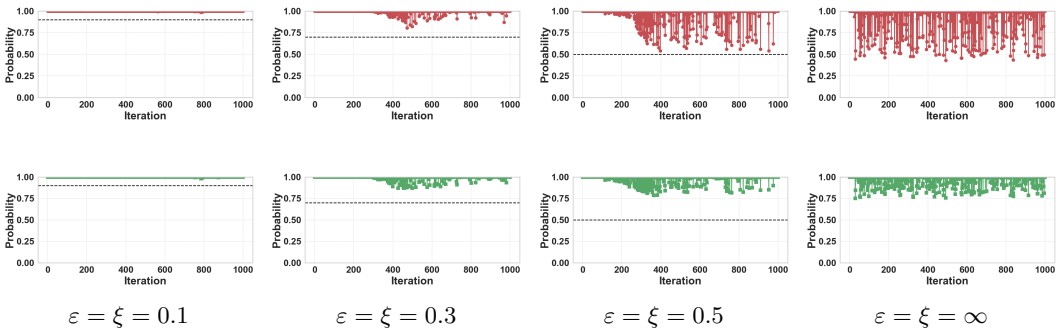

Figure 3: Safety (top row) and reset (bottom row) probabilities over iterations for various thresholds.

**Exploration rate and coverage.** In Figure 4, we plot the selected controls after 1000 iterations for various threshold pairs $\varepsilon = \xi$, with values increasing from left to right in $\{0.1, 0.3, 0.5, +\infty\}$. Figures 2 and 4 clearly illustrate the exploration-safety trade-off. Relaxing thresholds leads to broader control coverage and faster information gain, but with decreased safety guarantees. Conversely, strict thresholds restrict exploration, particularly around regions with safety or reset probabilities close to the specified thresholds. This aligns with the intuition supported by our theoretical analysis: sample complexity tends to increase in regions where smaller uncertainty is required to proceed safely. As a result, these regions act as bottlenecks, slowing down the process and potentially stopping exploration within the connected component that satisfies the constraints and includes the initial safe control. Additional results on information gain across iterations are provided in Appendix C.2.

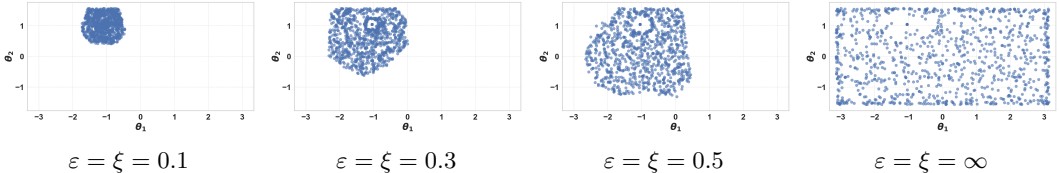

Figure 4: Control coverage for various thresholds.

**Safety and reset level prediction.** In Table 1, we quantify the accuracy of the learned model for various threshold pairs $\varepsilon = \xi$ in $\{0.1, 0.3, 0.5, +\infty\}$ by evaluating the prediction quality of the safety and reset levels over 1000 predictions. We report the mean squared error (MSE) and the standard deviation, with the ground truth provided by Monte Carlo estimates based on 100 samples (displayed in Figure 5). In Figure 6, we plot the learned safety and reset maps, whose accuracies can be qualitatively assessed by comparing their values with those in Figure 5. As expected, prediction accuracy improves as safety constraints are relaxed, due to the broader coverage of the control space.

Table 1: Safety and reset level prediction error statistics (MSE $\pm$ Std. Dev.)

| Model $(\varepsilon, \xi)$ | Safety MSE | Reset MSE |
|---|---|---|
| $(0.1, 0.1)$ | $0.7010 \pm 0.3847$ | $0.6919 \pm 0.3764$ |
| $(0.3, 0.3)$ | $0.5217 \pm 0.4254$ | $0.5197 \pm 0.4161$ |
| $(0.5, 0.5)$ | $0.3736 \pm 0.4299$ | $0.3701 \pm 0.4258$ |
| $(+\infty, +\infty)$ | $0.0023 \pm 0.0065$ | $0.0024 \pm 0.0062$ |

**Dynamics prediction.** To verify that our method captures the underlying system dynamics, we compare predicted trajectory densities with ground-truth trajectories under known-safe controls. Qualitative results show close agreement in both mean and variance. Full visualizations and evaluation details are provided in Appendix C.1.

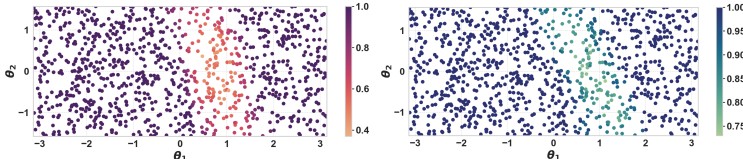

Figure 5: Ground-truth safety (left) and reset (right) probabilities estimated via 100 Monte Carlo samples for 1000 randomly selected controls.

**Computational considerations.** Our method runs end-to-end in under 32 minutes on standard hardware, covering candidate selection, simulation, evaluation, and model updates. Appendix B.4 provides runtimes, hardware specs, and potential optimizations (e.g., sketching, parallelization), confirming the method's practicality on standard hardware.

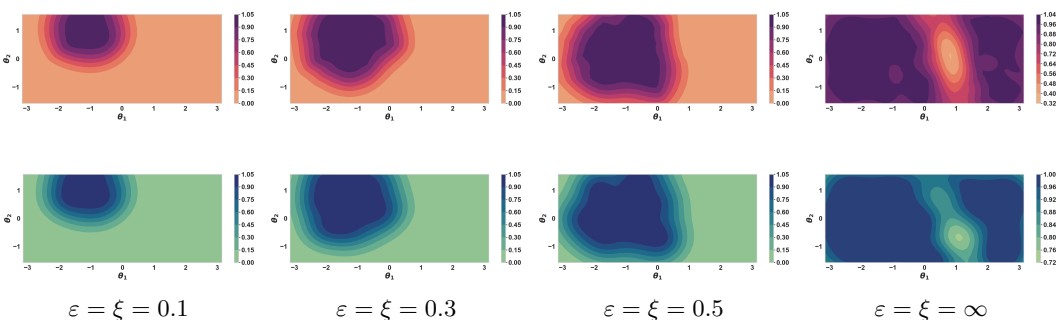

$$\varepsilon = \xi = 0.1 \qquad \varepsilon = \xi = 0.3 \qquad \varepsilon = \xi = 0.5 \qquad \varepsilon = \xi = \infty$$

Figure 6: Learned safety (top row) and reset (bottom row) level maps for various thresholds.

## 7 Conclusion

We introduced a provably safe and efficient method for learning controlled stochastic dynamics from trajectory data. By leveraging kernel-based confidence bounds and smoothness assumptions, our method incrementally expands an initial safe control set, ensuring that all trajectories remain within predefined safety regions throughout the learning process. Theoretical guarantees were established for both safety and estimation accuracy, with learning rates that adapt to the Sobolev regularity of the true dynamics. Numerical experiments corroborate our theoretical findings regarding safety and estimation accuracy. By tuning the safety ($\varepsilon$) and reset ($\xi$) thresholds, users can explicitly control the trade-off between conservative safety satisfaction and exploratory behavior. While our experimental validation focuses on a low-dimensional setting, the theoretical results scale with dimension: the convergence rates for the proposed estimators decrease polynomially with dimension and can mitigate the curse of dimensionality under sufficient smoothness. This makes the method applicable to higher-dimensional systems, which we plan to investigate in future work. Further research will include validation on physical systems (e.g., autonomous robots), improved scalability through fast kernel methods (e.g., sketching or incremental updates), comparisons with safe RL baselines, systematic analyses of kernel and threshold selection, and extensions to handle abrupt dynamics and non-diffusive disturbances such as jump processes arising in pedestrian–vehicle interactions and hybrid systems. These developments will further support applications in safety-critical control and decision-making under uncertainty.

## Acknowledgements

The Agence Nationale de la Recherche (grant ANR-22-CE48-0006, PI: R.B.) provided funds to assist the authors with their research. A.R. acknowledges support from the European Research Council (grant REAL 947908).

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

# A  Proofs

## A.1  Notations

We denote by $S_+^n$ the set of positive-definite matrices in $\mathbb{R}^{n \times n}$. The space of all measurable functions mapping a set $A$ to a set $B$ is denoted by $\mathcal{F}(A, B)$. For any vectors $u, v$, their tensor product is denoted by $u \otimes v$. For any conformable operator $A$, we define $A_\lambda \triangleq A + \lambda I$, where $I$ is the identity operator. The minimum and maximum between two scalars $a, b$ are denoted as $a \wedge b \triangleq \min(a, b)$ and $a \vee b \triangleq \max(a, b)$. To quantify function smoothness, we use Sobolev spaces. For a domain $\Omega \subset \mathbb{R}^d$, the Sobolev space $H^\nu(\Omega)$ consists of functions whose weak derivatives up to order $\nu$ exist and are square-integrable. The Sobolev embedding theorem states that if $\nu > d/2 + k$, then functions in $H^\nu(\Omega)$ are at least $C^k(\Omega)$-smooth. In our work, we consider domains such as $\mathbb{R}^n$ for spatial variables and $D \times [0, T_{\max}]$ for control-time spaces, where $D \subset \mathbb{R}^m$ is a compact set of control parameters. Finally, for the RKHS $\mathcal{G}$ on $D \times [0, T_{\max}]$ with kernel $k$, and any $p : D \times [0, T_{\max}] \times \mathbb{R}^n \to \mathbb{R}$, we define the mixed sup–norm $\|p\|_{L^\infty(\mathbb{R}^n; \mathcal{G})} \triangleq \sup_{x \in \mathbb{R}^n} \left\| (\theta, t) \mapsto p(\theta, t, x) \right\|_{\mathcal{G}}$.

## A.2  Proofs organization

The proofs are structured as follows.

1. **Validity of confidence intervals (Section A.3)**: First, we establish the validity of the confidence intervals.

2. **Safety and reset guarantees (Section A.4)**: We prove that controls chosen by the algorithm maintain safety and reset properties.

3. **Sample complexity bounds (Section A.5)**: Next, we analyze the sample complexity required to achieve desired accuracy.

4. **Learning rates for safety, reset, and density estimates at fixed $(\theta, t)$ (Section A.6):** We derive learning rates, under Sobolev regularity, for estimating the safety and reset levels, as well as the state density, evaluated at fixed $(\theta, t)$ pairs.

5. **Safe learning of controlled Sobolev dynamics (Section A.7)**: Finally, we establish complete guarantees for safely learning controlled dynamics with Sobolev regularity.

## A.3  Proof of the validity of confidence intervals

**Assumption (A4)** (Attainability). *There exists a bounded and continuous reproducing kernel $k$ defined on $D \times [0, T_{\max}]$, with associated RKHS $\mathcal{G}$, such that the safety and reset level functions satisfy*

$$s, r \in \mathcal{G}. \tag{20}$$

This assumption ensures that the safety and reset level functions can be represented in a suitable function space for estimation. In particular, it encodes prior knowledge; for example, if the functions are known to lie in a Sobolev (Hilbert) space $H^m$, one may choose a Sobolev kernel of order $m$. This is a standard assumption in the literature on kernel methods. In many practical applications, such as industrial process control and robotics [39; 51], SDEs with smooth coefficients produce smooth probability densities [17], ensuring that $s(\theta, t)$ is smooth and can be accurately represented by Gaussian or Sobolev kernels. Therefore, this assumption is mild in practice.

This lemma establishes the relevance of the defined confidence intervals.

**Lemma A.1** (Validity of confidence intervals). *Under Assumptions (A1), (A2), and (A4), for any $(\theta, t) \in D \times [0, T_{\max}]$ and $\lambda > 0$,*

$$|\hat{s}_N(\theta, t) - s(\theta, t)| \le \beta_N^s \sigma_N(\theta, t), \tag{21}$$

$$|\hat{r}_N(\theta, t) - r(\theta, t)| \le \beta_N^r \sigma_N(\theta, t), \tag{22}$$

*where* $\beta_N^s \triangleq \lambda^{-1} N^{-1/2} \max_{i \in [\![1, N]\!]} |\hat{s}_{\theta_i, t_i} - s(\theta_i, t_i)| + \|s\|_{\mathcal{G}}$, $\beta_N^r \triangleq \lambda^{-1} N^{-1/2} \max_{i \in [\![1, N]\!]} |\hat{r}_{\theta_i, t_i} - r(\theta_i, t_i)| + \|r\|_{\mathcal{G}}$.

*Proof.* Without loss of generality, we provide the detailed proof only for $s_N$, since the proof for $r_N$ is entirely analogous.

We start by recalling the definitions

$$\hat{s}_N(\theta, t) \triangleq \hat{S}^*(K + N\lambda I)^{-1} k(\theta, t), \tag{23}$$

$$\sigma_N^2(\theta, t) \triangleq k((\theta, t), (\theta, t)) - k(\theta, t)^*(K + N\lambda I)^{-1} k(\theta, t), \tag{24}$$

where $k(\theta) \triangleq (k((\theta, t), (\theta_i, t_i)))_{i=1}^N$, $K \triangleq (k((\theta_i, t_i), (\theta_j, t_j)))_{i,j=1}^N$, and $\hat{S} = (\hat{s}_{\theta_i, t_i})_{i=1}^N$.

We define the feature map $\phi : (\theta, t) \in D \times [0, T_{\max}] \mapsto k((\theta, t), \cdot) \in \mathcal{G}$, and the operators

$$\Phi \triangleq [\phi(\theta_1, t_1), \dots, \phi(\theta_N, t_N)] \in \mathcal{G} \otimes \mathbb{R}^N, \tag{25}$$

$$\hat{C} \triangleq \frac{1}{N} \sum_{i=1}^N \phi(\theta_i, t_i) \otimes \phi(\theta_i, t_i), \tag{26}$$

such that $\hat{C} = \frac{1}{N} \Phi \Phi^*$, $K = \Phi^* \Phi$, $k((\theta, t), (\theta', t')) = \langle \phi(\theta, t), \phi(\theta', t') \rangle_{\mathcal{G}}$, and $k(\theta, t) = \Phi^* \phi(\theta, t)$.

Then,

$$\hat{s}_N(\theta) \triangleq \hat{S}^*(K + N\lambda I_{\mathbb{R}^N \otimes \mathbb{R}^N})^{-1} k(\theta, t), \tag{27}$$

$$= \hat{S}^*(\Phi^* \Phi + N\lambda I_{\mathbb{R}^N \otimes \mathbb{R}^N})^{-1} \Phi^* \phi(\theta, t), \tag{28}$$

$$= \hat{S}^* \Phi^*(\Phi \Phi^* + N\lambda I_{\mathcal{G} \otimes \mathcal{G}})^{-1} \phi(\theta, t), \tag{29}$$

$$= N^{-1} \hat{S}^* \Phi^*(\hat{C} + \lambda I_{\mathcal{G} \otimes \mathcal{G}})^{-1} \phi(\theta, t), \tag{30}$$

using the push-through equality $(I + AB)^{-1} A = A(I + BA)^{-1}$ for any conformal operators $A, B$.

Moreover,

$$\sigma_N^2(\theta) \triangleq k((\theta, t), (\theta, t)) - k(\theta, t)^*(K + N\lambda I_{\mathbb{R}^N \otimes \mathbb{R}^N})^{-1} k(\theta, t) \tag{31}$$

$$= \phi(\theta, t)^*(I_{\mathcal{G} \otimes \mathcal{G}} - \Phi(\Phi^* \Phi + N\lambda I_{\mathbb{R}^N \otimes \mathbb{R}^N})^{-1} \Phi^*) \phi(\theta, t) \tag{32}$$

$$= \phi(\theta, t)^*(I_{\mathcal{G} \otimes \mathcal{G}} - N\hat{C}(N\hat{C} + N\lambda I_{\mathcal{G} \otimes \mathcal{G}})^{-1}) \phi(\theta, t) \tag{33}$$

$$= \lambda \phi(\theta, t)^*(\hat{C} + \lambda I_{\mathcal{G} \otimes \mathcal{G}})^{-1} \phi(\theta, t) \tag{34}$$

$$= \lambda \| \hat{C}_\lambda^{-1/2} \phi(\theta, t) \|_{\mathcal{G}}^2, \tag{35}$$

again using the push-through equality.

To bound the error $|\hat{s}_N(\theta, t) - s(\theta, t)|$, we decompose it as

$$|\hat{s}_N(\theta, t) - s(\theta, t)| \leq \underbrace{|\hat{s}_N(\theta, t) - s_N(\theta, t)|}_{\triangleq (A)} + \underbrace{|s_N(\theta, t) - s(\theta, t)|}_{\triangleq (B)}, \tag{36}$$

where $S \triangleq (s(\theta_i, t_i))_{i=1}^N$, and $s_N(\theta, t) \triangleq N^{-1} S^* \Phi^*(\hat{C} + \lambda)^{-1} \phi(\theta, t)$.

For the first term $(A)$, we have

$$(A) = N^{-1} |(\hat{S} - S)^* \Phi^* \hat{C}_\lambda^{-1} \phi(\theta, t)| \tag{37}$$

$$\leq N^{-1} \| \hat{S} - S \|_{\mathbb{R}^N} \| \Phi^* \hat{C}_\lambda^{-1} \phi(\theta, t) \|_{\mathcal{G}} \tag{38}$$

$$\leq N^{-1/2} \max_{i \in [\![1, N]\!]} |\hat{s}_{\theta_i, t_i} - s(\theta_i, t_i)| \, \| \hat{C}^{1/2} \hat{C}_\lambda^{-1} \phi(\theta, t) \|_{\mathcal{G}} \tag{39}$$

$$\leq N^{-1/2} \lambda^{-1} \max_{i \in [\![1, N]\!]} |\hat{s}_{\theta_i, t_i} - s(\theta_i, t_i)| \, \sigma_N(\theta, t). \tag{40}$$

From Assumption (A4), $s(\theta, t) = \langle s, \phi(\theta, t) \rangle_{\mathcal{G}}$, such that $S = \Phi^* s$, and then $s_N(\theta, t) = s^* \Phi \Phi^*(\hat{C} + \lambda)^{-1} \phi(\theta, t) = s^* \hat{C}(\hat{C} + \lambda)^{-1} \phi(\theta, t)$.

Therefore, similarly, for the second term $(B)$, we have

$$(B) = |s^*(\hat{C} \hat{C}_\lambda^{-1} - I) \phi(\theta, t)| \tag{41}$$

$$= \lambda |s^* \hat{C}_\lambda^{-1} \phi(\theta, t)| \tag{42}$$

$$\leq \lambda^{1/2} \| s \|_{\mathcal{G}} \| \hat{C}_\lambda^{-1/2} \phi(\theta, t) \|_{\mathcal{G}} \tag{43}$$

$$= \| s \|_{\mathcal{G}} \sigma_N(\theta, t). \tag{44}$$

Combining the bounds for $(A)$ and $(B)$, we obtain the bound for $s$. Similar proof yields the bound for $r$.

$\square$

In the following lemma, we derive confidence intervals for the proposed estimate of the system's dynamics $p : \theta, t, x \mapsto p_\theta(t, x)$.

**Lemma A.2.** *Under Assumptions (A1), (A2), and (A3), for any $(\theta, t) \in D \times [0, T_{\max}]$, we have*

$$\|\hat{p}_\theta(t, \cdot) - p_\theta(t, \cdot)\|_{L^\infty(\mathbb{R}^n)} \leq \beta_N^p \sigma_N(\theta, t), \tag{45}$$

*by defining $\beta_N^p \triangleq \lambda^{-1} N^{-1/2} \max_{i \in [\![1, N]\!]} \|\hat{p}_{\theta_i, t_i}(\cdot) - p_{\theta_i}(t_i, \cdot)\|_{L^\infty(\mathbb{R}^n)} + \|p\|_{L^\infty(\mathbb{R}^n; \mathcal{G})}$*

*Proof.* Given any $(\theta, t, x) \in D \times [0, T_{\max}] \times \mathbb{R}^n$, we have

$$\hat{p}_\theta(t, x) \triangleq \sum_{i=1}^{N} \alpha_i(\theta, t) \hat{p}_{\theta_i, t_i}(x), \tag{46}$$

with $\alpha(\theta, t) = (K + N\lambda I)^{-1} k(t, \theta)$, $K = (k((\theta_i, t_i), (\theta_j, t_j)))_{i,j=1}^{N}$, $k(\theta, t) = (k(\theta_i, t_i))_{i=1}^{N}$.

We define the feature map $\phi = (\theta, t) \in D \times [0, T_{\max}] \mapsto k((\theta, t), \cdot) \in \mathcal{G}$, and the operators

$$\Phi \triangleq [\phi(\theta_1, t_1), \ldots, \phi(\theta_N, t_N)] \in \mathcal{G} \otimes \mathbb{R}^N, \tag{47}$$

$$\hat{C} \triangleq \frac{1}{N} \sum_{i=1}^{N} \phi(\theta_i, t_i) \otimes \phi(\theta_i, t_i), \tag{48}$$

such that $\hat{C} = \frac{1}{N} \Phi \Phi^*$, $K = \Phi^* \Phi$, $k((\theta, t), (\theta', t')) = \langle \phi(\theta, t), \phi(\theta', t') \rangle_{\mathcal{G}}$, and $k(\theta, t) = \Phi^* \phi(\theta, t)$. With same derivations than in the proof of Lemma A.1, for any $(\theta, t, x) \in D \times [0, T_{\max}] \times \mathbb{R}^n$, we have

$$|\hat{p}_\theta(t, x) - p_\theta(t, x)| \leq (A) + (B) \tag{49}$$

with

$$(A) \leq \lambda^{-1} N^{-1/2} \max_{i \in [\![1, N]\!]} |\hat{p}_{\theta_i, t_i}(x) - p_{\theta_i}(t_i, x)| \, \sigma_N(\theta, t). \tag{50}$$

$$(B) \leq \|p_\cdot(\cdot, x)\|_{\mathcal{G}} \, \sigma_N(\theta, t). \tag{51}$$

Therefore, for any $(\theta, t, x) \in D \times [0, T_{\max}] \times \mathbb{R}^n$, we have

$$\|\hat{p}_\theta(t, \cdot) - p_\theta(t, \cdot)\|_{L^\infty(\mathbb{R}^n)} \leq \beta_N^p \sigma_N(\theta, t), \tag{52}$$

by defining $\beta_N^p \triangleq \lambda^{-1} N^{-1/2} \max_{i \in [\![1, N]\!]} \|\hat{p}_{\theta_i, t_i}(\cdot) - p_{\theta_i}(t_i, \cdot)\|_{L^\infty(\mathbb{R}^n)} + \|p\|_{L^\infty(\mathbb{R}^n; \mathcal{G})}$

$\square$

### A.4 Proof of safety and reset guarantees

A direct consequence of Lemma A.1 is the safety and reset guarantees for the method.

**Lemma A.3** (Safety guarantees). *Under Assumptions (A1), (A2), and (A4), the algorithm selects only safe trajectories. Namely, for any $i \in \mathbb{N}^*$, we have $s^\infty(\theta_i, T_i) \geq 1 - \varepsilon$.*

*Proof.* For any $i \in [\![1, N]\!]$, we have by construction $\inf_{t \in [0, T_i]} (\hat{s}_N(\theta_i, t) - \beta_N \sigma_N(\theta_i, t)) \geq 1 - \varepsilon$.

Moreover, from Lemma A.1, we have $s(\theta, t) \geq \hat{s}(\theta, t) - \beta_N \sigma_N(\theta, t)$ for any $(\theta, t) \in D \times [0, T_{\max}]$, such that

$$s^\infty(\theta_i, T_i) \triangleq \inf_{t \in [0, T_i]} s(\theta_i, t) \tag{53}$$

$$\geq \inf_{t \in [0, T_i]} (\hat{s}(\theta_i, t) - \beta_N \sigma_N(\theta_i, t)) \tag{54}$$

$$\geq 1 - \varepsilon. \tag{55}$$

$\square$

**Lemma A.4** (Reset guarantees). *Under Assumptions (A1), (A2), and (A4), the algorithm selects only resetting trajectories. Namely, for any $i \in \mathbb{N}^*$, we have $r(\theta_i, T_i) \geq 1 - \xi$.*

*Proof.* Similar proof as for Lemma A.3. □

## A.5 Proof of sample complexity of confidence intervals

**Assumption (A5)** (Sublinear information growth). *Let the maximum information gain up to $N$ observations be defined as*

$$\gamma_N \triangleq \max_{(\theta_i, t_i)_{i=1}^N \in (D \times [0, T_{\max}])^N} \frac{1}{2} \sum_{i=1}^N \log\left(1 + \frac{\sigma_i^2(\theta_i, t_i)}{\lambda N}\right). \tag{56}$$

*We assume there exist constants $\alpha \in [0, 1]$ and $c > 0$ such that, for all $N \in \mathbb{N}$,*

$$\gamma_N \leq cN^\alpha. \tag{57}$$

This assumption always holds for $\alpha = 1$ when $k$ is bounded, since $\gamma_N$ is a sum of terms bounded by $\kappa = \sup_{(\theta,t) \in D \times [0, T_{\max}]} k((\theta, t), (\theta, t))$. As $\alpha$ decreases within $[0, 1]$, the assumption becomes stricter. It quantifies the maximal total information that can be acquired about the unknown function after $N$ observations. Specifically, when $\alpha < 1$, the sublinear growth of information gain with $N$ reflects diminishing returns as more data points are observed. This growth rate measures the effective dimension of the learning problem, influenced by the regularity of the RKHS and the dimensionality of $D \times [0, T_{\max}]$; higher regularity of RKHS functions and lower dimensionality of $D \times [0, T_{\max}]$ lead to a faster decay in information gain.

**Example 2** (Sublinear growth for common kernels). *For many commonly used kernels, $\gamma_N$ exhibits sublinear growth in $N$, which is crucial for obtaining sublinear regret bounds in bandit problems. For example, assuming a compact domain $D \subset \mathbb{R}^m$ and setting $\lambda = N^{-1}$:*

- *RBF kernel: $\gamma_N = \mathcal{O}(\log^{m+1} N)$. This logarithmic growth arises from the high smoothness of the RBF kernel, which causes a rapid reduction in uncertainty about the unknown function as more observations are collected.*

- *Matérn kernel with $\nu > 1$: $\gamma_N = \mathcal{O}(N^{m/(m+2\nu)} \log^{2m/(m+2\nu)} N)$. This sublinear growth rate is faster than that of the RBF kernel, reflecting the lower smoothness of the Matérn kernel.*

*We refer the reader to Seeger et al. [59]; Srinivas et al. [54]; Vakili et al. [60] for additional examples and their associated proofs, which show that the growth rate of $\gamma_N$ depends on the eigenvalue decay in the Mercer expansion of the kernel, when available.*

This lemma establishes the sample complexity of the proposed confidence intervals.

**Lemma A.5** (Sample complexity of confidence intervals). *Under Assumptions (A1), (A2), and (A5), for any $\eta > 0$, considering the stopping condition $\max_{(\theta, t, T) \in \Gamma_N} \sigma_N(\theta, t) < \eta$, then the proposed method stops in $N = c\eta^{-\frac{2}{1-\alpha}}$ steps where $c > 0$ is a constant that does not depend on $N$ and $\eta$.*

*Proof.* Hence, under Assumption (A5), when the algorithm stops we have

$$N\eta^2 \leq \sum_{i=1}^N \sigma_i^2(\theta_i, t_i) \leq \frac{1}{2} \sum_{i=1}^N \log(1 + \sigma_i^2(\theta_i, t_i)) \leq \gamma_N \leq cN^\alpha, \tag{58}$$

using $x \leq 2\log(1 + x)$ for $x \in [0, 1]$, and overloading the constant $c > 0$. Hence, $N \leq c^{-1}\eta^{-\frac{2}{1-\alpha}}$. □

## A.6 Proof of learning rates for system maps estimation at fixed $(\theta, t)$

This lemma provides a bound on the density estimation error at the selected points $(\theta_i, t_i)$.

**Lemma A.6** (Density estimation learning rates). *For each $i \in [\![1, N]\!]$, define $p_i \triangleq p_{\theta_i}(t_i, \cdot)$, and*

$$\hat{p}_i \triangleq \hat{p}_{\theta_i, t_i} \triangleq \frac{1}{Q} \sum_{j=1}^{Q} \rho_R(x - X_{\theta_i}(w_j, t_i)), \tag{59}$$

*where $\rho_R(x) \triangleq R^{n/2} \|x\|^{-n/2} B_{n/2}(2\pi R\|x\|)$, $R > 0$, and $B_{n/2}$ is the Bessel J function of order $n/2$.*

*Under Assumptions (A1)-(A2), assume that $\sup_{(\theta, t) \in D \times [0, T_{\max}]} \|p_\theta(t, \cdot)\|_{H^\nu} < \infty$ for $\nu > n/2$. Set $R = Q^{\frac{1}{n+2\nu}}$. Then, with probability at least $1 - \delta$, the following holds*

$$\max_{i \in [\![1, N]\!]} \|\hat{p}_i - p_i\|_{L^\infty(\mathbb{R}^n)} \leq c \log(4N/\delta)^{1/2} Q^{\frac{n-2\nu}{2n+4\nu}}, \text{[2]} \tag{60}$$

*for some constant $c > 0$ independent of $N, Q, \delta$.*

*Proof.* Fix any $\varepsilon > 0$. The Sobolev embedding $H^{n/2+\varepsilon}(\mathbb{R}^n) \hookrightarrow L^\infty(\mathbb{R}^n)$ gives

$$\|\hat{p}_i - p_i\|_{L^\infty(\mathbb{R}^n)} \leq c_\varepsilon \|\hat{p}_i - p_i\|_{H^{n/2+\varepsilon}(\mathbb{R}^n)}, \qquad i = 1, \ldots, N,$$

where $c_\varepsilon > 0$ depends only on $n, \varepsilon$.

Following steps 3 and 4 of Theorem 5.3 in Bonalli and Rudi [17], for $\nu > n/2 + \varepsilon$ we establish

$$\max_{i \in [\![1, N]\!]} \|\hat{p}_i - p_i\|_{H^{n/2+\varepsilon}} \leq \max_{i \in [\![1, N]\!]} \|p_i\|_{H^\nu} R^{\frac{n}{2}+\varepsilon-\nu} \\ + 2^{n/2+\varepsilon} R^{\frac{n}{2}+\varepsilon} 3(V_n R)^{\frac{n}{2}} \log(4N/\delta)^{1/2} Q^{-1/2}. \tag{61}$$

where $V_n$ is the volume of the $n$-dimensional unit ball.

Under the assumption that $\sup_{(\theta, t) \in D \times [0, T_{\max}]} \|p_\theta(t, \cdot)\|_{H^\nu} < \infty$, we have in particular $\max_{i \in [\![1, N]\!]} \|p_i\|_{H^\nu} < \infty$. Hence, with probability at least $1 - \delta$,

$$\max_{i \in [\![1, N]\!]} \|\hat{p}_i - p_i\|_{H^{n/2+\varepsilon}} \leq c_\varepsilon \left( R^{\frac{n}{2}+\varepsilon-\nu} + R^{n+\varepsilon} \left[ \log(4N/\delta) \right]^{1/2} Q^{-1/2} \right), \tag{62}$$

for some $c_\varepsilon > 0$ independent of $N, Q, R, \delta$ (but possibly depending on $n, \nu, \varepsilon$).

Finally, setting $R = Q^{\frac{1}{n+2\nu}}$ balances the two terms in (62) and yields

$$\max_{i \in [\![1, N]\!]} \|\hat{p}_i - p_i\|_{H^{n/2+\varepsilon}} \leq c [\log(4N/\delta)]^{1/2} Q^{\frac{n-2\nu+2\varepsilon}{2n+4\nu}}, \tag{63}$$

for some constant $c > 0$ that does not depend on $N, Q, \delta$.

$\square$

This lemma bounds the pointwise estimation errors of the safety and reset levels at the selected points $(\theta_i, t_i)$ in terms of the $L^\infty$-norm error of the corresponding density estimations $\hat{p}_{\theta_i}(t_i, \cdot)$.

**Lemma A.7.** *Under Assumptions (A1)-(A2), we have*

$$\max_{i \in [\![1, N]\!]} |\hat{s}_{\theta_i, t_i} - s(\theta_i, t_i)| \leq V_s \max_{i \in [\![1, N]\!]} \|\hat{p}_i - p_i\|_{L^\infty}, \tag{64}$$

$$\max_{i \in [\![1, N]\!]} |\hat{r}_{\theta_i, t_i} - r(\theta_i, t_i)| \leq V_r \max_{i \in [\![1, N]\!]} \|\hat{p}_i - p_i\|_{L^\infty}, \tag{65}$$

*where $p_i \triangleq p_{\theta_i}(t_i, \cdot)$, $V_s \triangleq \int_{\mathbb{R}^n} \mathbb{1}_{\{g(x) \geq 0\}} dx$, and $V_r \triangleq \int_{\mathbb{R}^n} \mathbb{1}_{\{h(x) \geq 0\}} dx$.*

*Proof.* For any $i \in [\![1, N]\!]$, using Hölder's inequality, we have

$$|\hat{s}_{\theta_i, t_i} - s(\theta_i, t_i)| \leq V_s \|\hat{p}_i - p_i\|_{L^\infty}, \tag{66}$$

where $p_i \triangleq p_{\theta_i}(t_i, \cdot)$, and $V_s \triangleq \int_{\mathbb{R}^n} \mathbb{1}_{\{g(x) \geq 0\}} dx$.

Similarly, $|\hat{r}_{\theta_i, t_i} - r(\theta_i, t_i)| \leq V_r \|\hat{p}_i - p_i\|_{L^\infty}$ where $V_r \triangleq \int_{\mathbb{R}^n} \mathbb{1}_{\{h(x) \geq 0\}} dx$.

$\square$

---

[2]Throughout, exponents involving $n$ should be interpreted as $n + \varepsilon$ for arbitrarily small $\varepsilon > 0$.

## A.7 Proof of safe learning of controlled Sobolev dynamics

We conclude the following theorem from all previously established lemmas.

**Theorem A.8** (Safely learning controlled Sobolev dynamics)**.** *Let $\eta > 0$, and assume Assumptions (A1)–(A3) hold. Set $R = Q^{1/(n+2\nu)}$. Then there exist constants $c_1, \ldots, c_5 > 0$, independent of $N, Q, \delta, \eta$, such that if*

$$c_1 \log(4N/\delta)^{1/2} Q^{\frac{n-2\nu}{2n+4\nu}} \leq N^{-1/2},$$

*then the stopping condition $\max_{(\theta,t,T)\in\Gamma_N} \sigma_N(\theta,t) < \eta$ is satisfied after at most $N \leq c_2 \eta^{-2/(1-\alpha)}$ iterations for any $\alpha > (m+1)/(m+1+2\nu)$. Moreover:*

- **(Safety):** *All selected triples $(\theta_i, t_i, T_i)$ satisfy $s^\infty(\theta_i, T_i) \geq 1 - \varepsilon$ and $r(\theta_i, T_i) \geq 1 - \xi$. The final set $\Gamma_N$ includes only controls meeting these thresholds and can thus serve as a certified safe set for deployment.*

- **(Estimation accuracy):** *For all $(\theta, t, T) \in \Gamma_N$,*

$$\|\hat{p}_\theta(t,\cdot) - p_\theta(t,\cdot)\|_\infty \leq c_3 \eta, \quad |\hat{s}_N(\theta,t) - s(\theta,t)| \leq c_4 \eta, \quad |\hat{r}_N(\theta,t) - r(\theta,t)| \leq c_5 \eta.$$

*Proof.* From Lemma A.1 and Lemma A.2, for any $(\theta, t) \in D \times [0, T_{\max}]$ and $\lambda > 0$,

$$|\hat{s}_N(\theta,t) - s(\theta,t)| \leq \beta_N^s \sigma_N(\theta,t), \tag{67}$$

$$|\hat{r}_N(\theta,t) - r(\theta,t)| \leq \beta_N^r \sigma_N(\theta,t), \tag{68}$$

$$\|\hat{p}_\theta(t,\cdot) - p_\theta(t,\cdot)\|_{L^\infty(\mathbb{R}^n)} \leq \beta_N^p \sigma_N(\theta,t), \tag{69}$$

where $\beta_N^s \triangleq \lambda^{-1} N^{-1/2} \max_{i\in[\![1,N]\!]} |\hat{s}_{\theta_i,t_i} - s(\theta_i,t_i)| + \|s\|_{\mathcal{G}}$, $\beta_N^r \triangleq \lambda^{-1} N^{-1/2} \max_{i\in[\![1,N]\!]} |\hat{r}_{\theta_i,t_i} - r(\theta_i,t_i)| + \|r\|_{\mathcal{G}}$, and $\beta_N^p \triangleq \lambda^{-1} N^{-1/2} \max_{i\in[\![1,N]\!]} \|\hat{p}_{\theta_i,t_i}(\cdot) - p_{\theta_i}(t_i,\cdot)\|_{L^\infty(\mathbb{R}^n)} + \|p\|_{L^\infty(\mathbb{R}^n;\mathcal{G})}$.

Then, from Lemma A.7, we have

$$\max_{i\in[\![1,N]\!]} |\hat{s}_{\theta_i,t_i} - s(\theta_i,t_i)| \leq V_s \max_{i\in[\![1,N]\!]} \|\hat{p}_i - p_i\|_{L^\infty}, \tag{70}$$

$$\max_{i\in[\![1,N]\!]} |\hat{r}_{\theta_i,t_i} - r(\theta_i,t_i)| \leq V_r \max_{i\in[\![1,N]\!]} \|\hat{p}_i - p_i\|_{L^\infty}, \tag{71}$$

However, from Lemma A.6, with probability at least $1 - \delta$, we have

$$\max_{i\in[\![1,N]\!]} \|\hat{p}_i - p_i\|_{L^\infty(\mathbb{R}^n)} \leq c \log(4N/\delta)^{1/2} Q^{\frac{n-2\nu}{2n+4\nu}}, \tag{72}$$

such that there exists $c_1 > 0$ independent of $N, Q, \delta, \eta$ such that if

$$c_1 \log(4N/\delta)^{1/2} Q^{\frac{n-2\nu}{2n+4\nu}} \leq (V_s \vee V_r)^{-1} N^{-1/2}, \tag{73}$$

then

$$\max_{i\in[\![1,N]\!]} \|\hat{p}_i - p_i\|_{L^\infty(\mathbb{R}^n)} \leq N^{-1/2}. \tag{74}$$

Therefore, from Lemma A.7, we have, if $\lambda = N^{-1}$, for any $i \in [\![1, N]\!]$,

$$\beta_N^s \triangleq N^{1/2} \max_{j\in[\![1,N]\!]} |\hat{s}_{\theta_j,t_j} - s_{\theta_j,t_j}| + \|s\|_{\mathcal{G}} \tag{75}$$

$$\leq 1 + \|s\|_{\mathcal{G}}. \tag{76}$$

Similarly, we have $\beta_N^r \leq 1 + \|r\|_{\mathcal{G}}$ and $\beta_N^p \leq 1 + \|p\|_{L^\infty(\mathbb{R}^n;\mathcal{G})}$.

Assumption (A3) simultaneously guarantees $\|p\|_{L^\infty(\mathbb{R}^n;\mathcal{G})} \triangleq \sup_{x\in\mathbb{R}^n} \| (\theta,t) \mapsto p(\theta,t,x)\|_{\mathcal{G}} < +\infty$, $\sup_{(\theta,t)\in D\times[0,T_{\max}]} \|p_\theta(t,\cdot)\|_{H^\nu} < \infty$ with $\nu > n/2$. Furthermore, considering a kernel that induces a Sobolev RKHS $\mathcal{G}$ of order $\nu$, Assumption (A4) holds true, and Assumption (A5) holds for any $\alpha > (m+1)/(m+1+2\nu)$ as mentioned in the examples of Assumption (A5).

Therefore, from Lemma A.3, and Lemma A.4, we have that:

- All selected controls are safe, i.e., $s^\infty(\theta_i, T_i) \geq 1 - \varepsilon$ for any $i \in \mathbb{N}^*$.

- All selected controls ensure reset, i.e., $r(\theta_i, T_i) \geq 1 - \xi$ for any $i \in \mathbb{N}^*$.

Moreover, Lemma A.5 ensures that for any $\alpha > (m+1)/(m+1+2\nu)$, with probability at least $1 - \delta$, the stopping condition

$$\max_{\theta, t, T \in \Gamma_N} \sigma_N(\theta, t) < \eta, \tag{77}$$

is reached for $N \leq c_2 \eta^{-\frac{2}{1-\alpha}}$ where $c_2$ does not depend on $N, Q, \delta, \eta$.

$\square$

## B  Implementation details

This section details the implementation of the method described in Section 4, available on GitHub (lmotte/dynamics-safe-learn) as an open-source Python library. We detail all computational steps, including vectorized implementations using Python libraries such as NumPy for efficiency. Additionally, we outline the computational complexity of each step to provide insights into scalability.

### B.1  System estimation

**Density estimation.**    For each data point $(\theta_i, t_i, T_i)$, the density is estimated with

$$\hat{p}_{\theta_i, t_i}(x) = \frac{1}{Q} \sum_{j=1}^{Q} \rho_R(x - X_{u_{\theta_i}}(w_j^i, t_i)),$$

where $Q$ is the number of samples generated for each control $\theta_i$ and time $t_i$, $\rho_R(x)$ is a kernel density function, typically defined as $\rho_R(x) = R^{n/2} \|x\|^{-n/2} J_{n/2}(2\pi R \|x\|)$, with $R > 0$ and $J_{n/2}$ the Bessel function of order $n/2$, $X_{u_{\theta_i}}(w_j^i, t_i)$ are the system trajectories generated under control $u_{\theta_i}$.

**Computational complexity.**    Evaluating $\hat{p}_{\theta_i, t_i}(x)$ requires $\mathcal{O}(Q)$ operations per data point. This step is trivially parallelizable across data points, since all kernel evaluations are independent.

---

**Algorithm 1:** DensityEstimation($\{x_j\}_{j=1}^{Q}, \rho_R$)

---

**Input:** $Q$ trajectory samples $\{x_j\}_{j=1}^{Q}$ at $(\theta_i, t_i)$, kernel $\rho_R$
**Output:** density estimator $\hat{p}_{\theta_i, t_i} : \mathbb{R}^n \to \mathbb{R}_{\geq 0}$

1  **if** $x$ *is queried* **then  return** $\hat{p}_{\theta_i, t_i}(x) \leftarrow \dfrac{1}{Q} \sum_{j=1}^{Q} \rho_R(x - x_j)$

---

**Probability Computation** $(\hat{s}, \hat{r})$.    The vectors $\hat{P}$, $\hat{S}$, and $\hat{R}$ are constructed from the observed data points $\{(\theta_i, t_i, T_i)\}_{i=1}^{N}$ as follows:

$$\hat{P}(\cdot) = \begin{bmatrix} \hat{p}_{\theta_1}(t_1, \cdot) \\ \hat{p}_{\theta_2}(t_2, \cdot) \\ \vdots \\ \hat{p}_{\theta_N}(t_N, \cdot) \end{bmatrix}, \quad \hat{S} = \begin{bmatrix} \hat{s}_{\theta_1, t_1} \\ \hat{s}_{\theta_2, t_2} \\ \vdots \\ \hat{s}_{\theta_N, t_N} \end{bmatrix}, \quad \hat{R} = \begin{bmatrix} \hat{r}_{\theta_1, t_1} \\ \hat{r}_{\theta_2, t_2} \\ \vdots \\ \hat{r}_{\theta_N, t_N} \end{bmatrix}.$$

The vectors $\hat{S}$ and $\hat{R}$ are computed by integrating the densities over the safe and reset regions, respectively:

$$\hat{s}_{\theta_i, t_i} = \int_{\{x \in \mathbb{R}^n : g(x) \geq 0\}} \hat{p}_{\theta_i, t_i}(x) \, dx,$$

$$\hat{r}_{\theta_i, t_i} = \int_{\{x \in \mathbb{R}^n : h(x) \geq 0\}} \hat{p}_{\theta_i, t_i}(x) \, dx.$$

These integrals are approximated using Monte Carlo integration. If we can sample $x_k \sim \hat{p}_{\theta_i, t_i}$, then

$$\hat{s}_{\theta_i, t_i} \approx \frac{1}{Q'} \sum_{k=1}^{Q'} \mathbb{1}\{g(x_k) \geq 0\}, \qquad \hat{r}_{\theta_i, t_i} \approx \frac{1}{Q'} \sum_{k=1}^{Q'} \mathbb{1}\{h(x_k) \geq 0\}.$$

Alternatively, using trajectory samples $\{X_{u_{\theta_i}}(w_j^i, t_i)\}_{j=1}^Q$,

$$\hat{s}_{\theta_i, t_i} \approx \frac{1}{Q} \sum_{j=1}^Q \mathbb{1}\{g(X_{u_{\theta_i}}(w_j^i, t_i)) \geq 0\}, \qquad \hat{r}_{\theta_i, t_i} \approx \frac{1}{Q} \sum_{j=1}^Q \mathbb{1}\{h(X_{u_{\theta_i}}(w_j^i, t_i)) \geq 0\}.$$

Here $\mathbb{1}\{\cdot\}$ denotes the indicator function.

**Computational complexity.** Constructing $\hat{P}$ involves $N$ density evaluations, each costing $\mathcal{O}(Q)$, for a total of $\mathcal{O}(NQ)$. For $\hat{S}$ and $\hat{R}$, Monte Carlo based on samples from $\hat{p}$ costs $\mathcal{O}(NQ')$, while using trajectory samples costs $\mathcal{O}(NQ)$. Overall, constructing $(\hat{P}, \hat{S}, \hat{R})$ costs $\mathcal{O}(NQ)$ (trajectory MC) or $\mathcal{O}(N(Q + Q'))$ (using both). All computations are trivially parallelizable across evaluation points.

---

**Algorithm 2:** ComputeProbabilities($\hat{p}_{\theta_i, t_i}$, $g$, $h$, $Q'$, `mode`)

**Input:** density estimator $\hat{p}_{\theta_i, t_i}$, constraint functions $g, h$, number of MC samples $Q'$, mode $\in \{\texttt{density}, \texttt{trajectory}\}$

**Output:** probabilities $(\hat{s}_{\theta_i, t_i}, \hat{r}_{\theta_i, t_i})$

1 **if** `mode` $=$ `density` **then**

2 $\quad$ Sample $\{x_k\}_{k=1}^{Q'}$ **from** $\hat{p}_{\theta_i, t_i}$

3 $\quad \hat{s} \leftarrow \dfrac{1}{Q'} \sum_{k=1}^{Q'} \mathbf{1}\{g(x_k) \geq 0\}$

4 $\quad \hat{r} \leftarrow \dfrac{1}{Q'} \sum_{k=1}^{Q'} \mathbf{1}\{h(x_k) \geq 0\}$

5 **else** $\hfill$ `// trajectory`

6 $\quad$ Reuse trajectory samples $\{x_j\}_{j=1}^Q$ at $(\theta_i, t_i)$

7 $\quad \hat{s} \leftarrow \dfrac{1}{Q} \sum_{j=1}^Q \mathbf{1}\{g(x_j) \geq 0\}$

8 $\quad \hat{r} \leftarrow \dfrac{1}{Q} \sum_{j=1}^Q \mathbf{1}\{h(x_j) \geq 0\}$

9 **return** $(\hat{s}, \hat{r})$

---

**Kernel-based estimation of system maps.** Given $N$ observed data points $\{(\theta_i, t_i, T_i)\}_{i=1}^N$, the Gram matrix $K$ is constructed as

$$K_{ij} = k((\theta_i, t_i), (\theta_j, t_j)),$$

where $k$ is the kernel function. For a new input $(\theta, t)$, the estimates for the system dynamics, safety, and reset functions are

$$\hat{p}_\theta(t, x) = \hat{P}(x)(K + N\lambda I)^{-1} k(\theta, t),$$

$$\hat{s}_N(\theta, t) = \hat{S}(K + N\lambda I)^{-1} k(\theta, t),$$

$$\hat{r}_N(\theta, t) = \hat{R}(K + N\lambda I)^{-1} k(\theta, t),$$

where $k(\theta, t) = [k((\theta, t), (\theta_i, t_i))]_{i=1}^N$, $\lambda$ is a regularization parameter, $\hat{P}(\cdot) \triangleq (\hat{p}_{\theta_i, t_i}(\cdot))_{i=1}^N$, $\hat{S} = (\hat{s}_{\theta_i, t_i})_{i=1}^N$, $\hat{R} = (\hat{r}_{\theta_i, t_i})_{i=1}^N$.

The predictive uncertainty for $(\theta, t)$ is computed as

$$\sigma_N^2(\theta, t) = k((\theta, t), (\theta, t)) - k(\theta, t)^*(K + N\lambda I)^{-1} k(\theta, t).$$

**Computational complexity.** Gram matrix construction requires $\mathcal{O}(N^2)$ operations, matrix inversion involves $\mathcal{O}(N^3)$ operations. Evaluating $\hat{p}_N(\theta, t)$, $\hat{s}_N(\theta, t)$, $\hat{r}_N(\theta, t)$, or $\sigma_N^2(\theta, t)$, costs $\mathcal{O}(N^2)$ per prediction.

---

**Algorithm 3:** FitKernelMaps($\{(\theta_i, t_i)\}_{i=1}^N$, $\hat{P}(\cdot)$, $\hat{S}$, $\hat{R}$, $k$, $\lambda$)

**Input:** inputs $\{(\theta_i, t_i)\}_{i=1}^N$, targets $\hat{P}(\cdot)$, $\hat{S}$, $\hat{R}$, kernel $k$, regularization $\lambda$
**Output:** query operators for $\hat{p}_\theta(t, \cdot)$, $\hat{s}_N(\theta, t)$, $\hat{r}_N(\theta, t)$, $\sigma_N^2(\theta, t)$

1 Build $K$ with $K_{ij} \leftarrow k((\theta_i, t_i), (\theta_j, t_j))$; set $K_\lambda \leftarrow K + N\lambda I$
2 Compute and store $K_\lambda^{-1}$
3 **Precompute:** $\alpha_S \leftarrow K_\lambda^{-1} \hat{S}$, $\quad \alpha_R \leftarrow K_\lambda^{-1} \hat{R}$
4 **Query** at $(\theta, t)$ (and optionally at $x$):
    1. $k \leftarrow [\, k((\theta, t), (\theta_i, t_i)) \,]_{i=1}^N$
    2. $z \leftarrow K_\lambda^{-1} k$                         `// used for variance and density map`
    3. $\hat{s}_N(\theta, t) \leftarrow k^\top \alpha_S$
    4. $\hat{r}_N(\theta, t) \leftarrow k^\top \alpha_R$
    5. $\sigma_N^2(\theta, t) \leftarrow k((\theta, t), (\theta, t)) - k^\top z$
    6. **If density at $x$ is requested:** build $\hat{P}(x) \leftarrow [\hat{p}_{\theta_1, t_1}(x), \ldots, \hat{p}_{\theta_N, t_N}(x)]^\top$ and return
       $\hat{p}_\theta(t, x) \leftarrow \hat{P}(x)^\top z$

---

## B.2 Safe sampling

**Feasibility criteria.** The feasibility criteria for safe sampling are defined as:

$$\mathrm{LCB}_N^s(\theta, T) \triangleq \inf_{t \in [0, T]} \left( \hat{s}_N(\theta, t) - \beta_N^s \sigma_N(\theta, t) \right),$$

$$\mathrm{LCB}_N^r(\theta, T) \triangleq \hat{r}_N(\theta, T) - \beta_N^r \sigma_N(\theta, T),$$

where $\beta_N^s, \beta_N^r > 0$ are confidence parameters. The safe-resettable feasible set is:

$$\Gamma_N = \left\{ (\theta, t, T) \in D \times [0, T_{\max}]^2 \,\middle|\, t \leq T, \ \mathrm{LCB}_N^s(\theta, T) \geq 1 - \varepsilon, \ \mathrm{LCB}_N^r(\theta, T) \geq 1 - \xi \right\} \cup \Gamma_0,$$

with

$$\Gamma_0 = \left\{ (\theta, t, T_{\max}) \in D \times [0, T_{\max}]^2 \,\middle|\, (\theta, t) \in S_0 \cap R_0 \right\}.$$

**Sampling rule.** The next control, time, and trajectory are selected by solving:

$$(\theta_{N+1}, t_{N+1}, T_{N+1}) = \underset{(\theta, t, T) \in \Gamma_N}{\arg\max} \ \sigma_N(\theta, t),$$

where

$$\sigma_N^2(\theta, t) = k((\theta, t), (\theta, t)) - k(\theta, t)^*(K + N\lambda I)^{-1} k(\theta, t).$$

**Stopping rule.** Exploration terminates when:

$$\max_{(\theta, t, T) \in \Gamma_N} \sigma_N(\theta, t) < \eta,$$

where $\eta > 0$ is the user-defined uncertainty threshold.

**Computational complexity.**

- **Set construction ($\Gamma_N$):** Evaluating $\text{LCB}_N^s$ and $\text{LCB}_N^r$ involves $\mathcal{O}(N^2)$ operations for each candidate point. For $M$ candidate points, constructing $\Gamma_N$ costs $\mathcal{O}(MN^2)$.

- **Uncertainty evaluation:** Evaluating $\sigma_N$ involves matrix-vector multiplications and inversions, where Gram matrix construction costs $\mathcal{O}(N^2)$, matrix inversion costs $\mathcal{O}(N^3)$, and per-query uncertainty evaluation costs $\mathcal{O}(N^2)$.

- **Optimization ($\arg\max$):**
    - For discretization over $M$ candidates: $\mathcal{O}(MN^2)$,
    - For gradient-based optimization: $\mathcal{O}(kN^2)$, where $k$ is the number of optimization iterations, using a nonlinear constrained solver (e.g. SQP or interior-point). Each iteration involves computing the gradient of $\sigma_N(\theta, t)$, costing $\mathcal{O}(N^2)$.

**Efficient sampling algorithm.** In practice, the sampling process can be improved to reduce computational costs by focusing on promising regions and avoiding unnecessary evaluation of low-uncertainty points: (1) threshold-based filtering: select any candidate where $\sigma_N(\theta, t) > \eta$ to avoid costly global optimization while maintaining guarantees; (2) exclude evaluated points: skip candidates where $\sigma_N(\theta, t) < \eta$, assuming the uncertainty is non-increasing (true for $\lambda = 1/N$); (3) localized sampling: restrict $\Gamma_N$ to points near the initial safe set and previously selected points. Algorithm 4 implements a region-growing strategy that encourages local exploration around previously selected safe points.

---

**Algorithm 4:** Efficient sampling algorithm

**Input:** Initial safe-resettable set $S_0 \cap R_0$, threshold $\eta$, feasible set $\Gamma_N$
**Output:** Updated sets $\mathcal{P}_k$, $\mathcal{A}_k$, and the selected candidate (if found)

**1 Initialization:**
**2** Set $\mathcal{P}_0 = S_0 \cap R_0$ and $\mathcal{A}_0 = \emptyset$
**3** Define the feasible set using a localized search region:

$$\Gamma^k = \Gamma_N \cap \{(\theta, t, T) \mid d((\theta, t), \mathcal{P}_k) \leq r_k\} \setminus \mathcal{A}_k,$$

where $d((\theta, t), \mathcal{P}_k)$ denotes the minimum Euclidean distance from $(\theta, t)$ to any point in $\mathcal{P}_k$.

**4 Iterations:**
**5 for** $k = 0, 1, 2, \ldots$ **do**
**6**     **foreach** *candidate* $(\theta, t, T) \in \Gamma^k$ **do**
**7**         **if** $\sigma_N(\theta, t) > \eta$ **then**
**8**             Select the candidate $(\theta, t, T)$
**9**             Update the safe-resettable set:

$$\mathcal{P}_{k+1} = \mathcal{P}_k \cup \{(\theta, t, T)\}$$

            **return** $(\mathcal{P}_{k+1}, \mathcal{A}_k, (\theta, t, T))$
**10**         **end**
**11**         **else**
**12**             Update the excluded set:

$$\mathcal{A}_{k+1} = \mathcal{A}_k \cup \{(\theta, t, T)\}$$

**13**         **end**
**14**     **end**
**15**     Expand the radius: $r_k \to r_{k+1}$
**16**     Recompute $\Gamma^k$
**17**     **if** $\Gamma^k = \emptyset$ **then**
**18**         **return** $(\mathcal{P}_k, \mathcal{A}_k, \text{None})$
**19**     **end**
**20 end**

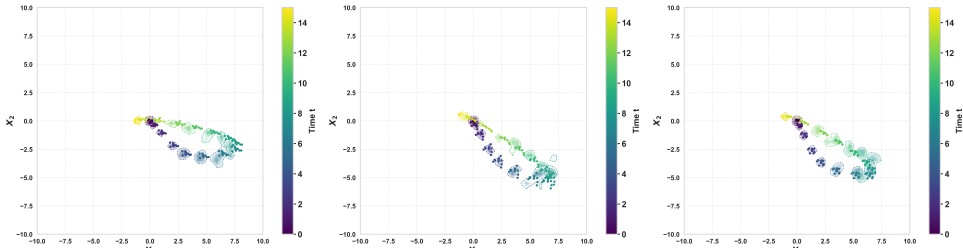

Figure 7: Predicted probability density of the trained model ($\varepsilon = \xi = 0.1$) along with 10 true trajectories, for three test controls ((-0.81 1.20), (-1.07 0.92), (-1.13 1.18)) from the known set of safe controls for this model. Predictions were computed on a spatial grid of 2500 points and times $t \in [\![0, 16]\!]$.

### B.3    Role and tuning of hyperparameters

Hyperparameters critically influence the balance between exploration, safety, and computational efficiency in our method. The safety and reset thresholds ($\varepsilon, \xi$) directly control this balance: lower thresholds enforce stricter constraints, restricting exploration to safer regions at the expense of slower coverage, whereas higher thresholds allow broader exploration but increase risk. The confidence parameters ($\beta_s, \beta_r$) modulate how conservatively the safe-resettable set expands, reflecting tolerance to uncertainty in safety and reset predictions. The parameters $\lambda$ and $\gamma$ define the smoothness of the estimated functions, thus capturing prior knowledge about the system dynamics. Different hyperparameters may be chosen for each map: $\gamma_{\text{kde}}$ and $\lambda_{\text{kde}}$ for density estimation, and $\gamma_{\text{collect}}, \lambda_{\text{collect}}, \beta_{\text{collect}}$ for safety and reset predictions, as these functions may have different smoothness characteristics. Typically, hyperparameters are tuned using validation data. Parameters for density estimation ($\gamma_{\text{kde}}, \lambda_{\text{kde}}$) can be optimized after data collection by maximizing log-likelihood. In contrast, the parameters governing safety and reset exploration ($\gamma_{\text{collect}}, \lambda_{\text{collect}}, \beta_{\text{collect}}$) must be set beforehand, as they directly impact safe exploration. When prior knowledge is limited, we recommend initially conservative settings—high $\gamma$, low $\lambda$, high $\beta$, and large kernel bandwidth $R$—and gradually relaxing them based on data-driven insights. In our experiments, $\gamma_{\text{kde}}$ and $\lambda_{\text{kde}}$ were visually tuned using validation controls $(2\pi/3, -\pi/3)$, $(-2\pi/3, -\pi/3)$, and $(0, -\pi/3)$. Meanwhile, $\gamma_{\text{collect}}, \lambda_{\text{collect}}, \beta_{\text{collect}}$ were set heuristically, assuming reasonable prior smoothness estimates to minimize computational overhead. Without such prior knowledge, comprehensive hyperparameter tuning would likely demand significantly higher computational resources, as extensive parameter searches become necessary.

### B.4    Computational considerations

To provide practical insight into computational requirements, we report measured execution times from experiments on a standard machine (Apple M3 Pro, 18GB RAM). Each training iteration required approximately $1.92s \pm 0.02$, totaling around $31.77 \pm 0.38$ minutes for 1000 iterations, based on 20 repeated runs. This includes candidate selection, trajectory simulation, computing safety and reset probabilities, and model updates at each iteration. Density predictions took approximately 10 seconds in average for computing $p(\theta, t, x)$ over a grid of 2500 $x$ for each considered $(\theta, t)$. Although the computational times are non-trivial, they remain manageable on standard hardware for the problem sizes considered. It should be noted that several approximation methods—such as sketching for matrix inversion and online matrix inversion—as well as parallelization techniques (e.g., parallelizing the simulations) can be leveraged to alleviate the computational burden. However, exploring these techniques is beyond the scope of this paper.

## C    Additional experimental results

### C.1    Dynamics prediction accuracy

In Figure 7, we present the predicted probability density from the model trained with $\varepsilon = \xi = 0.1$, alongside 10 true trajectories for three test controls ((-0.81 1.20), (-1.07 0.92), (-1.13 1.18)) chosen from the known set of safe controls with uncertainty below 0.1. The density is evaluated over a spatial grid of 2500 points and time steps $t \in [\![0, 16]\!]$. This visualization provides a qualitative assessment of

the model's dynamic prediction accuracy. The predicted probability distributions closely match the true dynamics, exhibiting similar means and variances over time.

## C.2   Information gain over iterations

To complement the analysis of exploration behavior, we report the cumulative information gain over the course of training, for different safety and reset thresholds $\varepsilon = \xi \in \{0.1, 0.3, 0.5, +\infty\}$. Figure 8 shows how the information gain evolves as new trajectory data is collected.

We observe that larger thresholds, which allow more aggressive exploration, result in faster information acquisition. In contrast, stricter thresholds slow down exploration and yield more gradual information growth. This reflects the fundamental trade-off between exploration and safety: ensuring high-probability safety requires restricting the sampling space, particularly in regions with high model uncertainty.

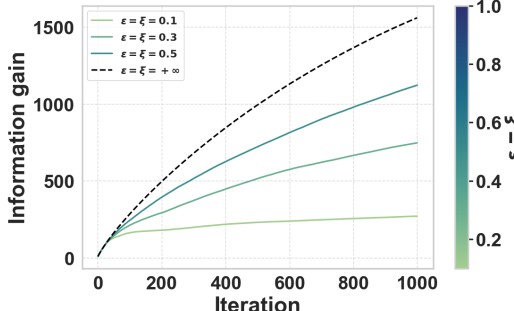

Figure 8:  Cumulative information gain over iterations for various thresholds $\varepsilon = \xi \in \{0.1, 0.3, 0.5, +\infty\}$.

