# OpenReview forum: "Safely Learning Controlled Stochastic Dynamics"
_NeurIPS.cc/2025/Conference — NeurIPS 2025 poster_

### Official Review · Reviewer_1MU6 · 2025-07-01

**Clarity:** 2
**Significance:** 3
**Originality:** 3
**Rating:** 5
**Confidence:** 4

**Summary:**

This paper presents a method for safely learning the dynamics of a nonlinear continuous time system, where safety is defined as high probability constraint satisfaction at each instant and learning is in terms of the state probability density map. Under assumptions on known control inputs for ensuring safety and resets, and on the smoothness of the (unknown) dynamics, the authors provide a method which guarantees safety and learning. The method uses kernel density estimation and ridge regression to estimate safety/reset/state density functions and their uncertainties, and safety constrained uncertainty maximization for sample new controllers.

**Questions:**

- Does this method suffer from a curse of dimensionality? Would it be feasible to present numerical experiments on systems with higher state dimensions or input parameters?
- In assumptions A1 and A2, is the requirement for all theta, or there exists a theta? The former is very restrictive but appears at odds with the written text.
- Can you expand more on how smoothness of dynamics relates to drift and diffusion terms?
- It is hard to parse the definition of initial safe resettable set, can you explain the role of t and t'?
- At the beginning of section 4.2, should it be index i instead of N? Or can you explain the role of these two indices?
- Can you further explain the assumption on N in theorem 5.1? It seems unintuitive to upper bound N by assumption. But maybe its actually better understood as a bound on Q, the number of samples per trajectory?
- Can you compare the size of the learned safe region to that of the known dynamics? If not theoretically, then at least empirically?
- Why is the definition of safety meaningful? Shouldn't we care about absolute safety integrated over time, rather than just at each instant probabilistically?

**Ethical Concerns:**

["NO or VERY MINOR ethics concerns only"]

**Final Justification:**

Author response addressed several of my concerns.

**Limitations:**

yes

**Quality:**

2

**Strengths And Weaknesses:**

Strengths:
1. The paper considers an important and difficult problem, and tackles challenges related to handling safety in continuous time stochastic dynamics.
2. The method comes with nice guarantees about about safety, accuracy, and sample complexity.
3. The method's performance is demonstrated in numerical experiments which illustrate the effect of different parameters.

Weaknesses:
1. There are some important technical details that are not clear, especially in the assumptions (see questions)
2. It seems that this method would suffer from a curse of dimensionality in the state or control parameter.
3. The guarantees do not address the size of the feasible set and the definition of safety is somewhat weak.
4. The text in the figures is too small. Additionally, it would be better to label the plots directly with the value of epsilon rather than in the caption.

---

> ### Author Rebuttal · Authors · 2025-07-30
>
> Thank you for engaging with the technical details. Your questions raised important clarifications that helped us improve the overall exposition.
>
> **(General comment).** This work takes a first step toward provably safe learning under unknown stochastic dynamics, a setting largely unexplored compared to prior work assuming known dynamics. Our focus is theoretical and methodological; code will be released for reproducibility. Experiments on more complex problems are part of ongoing work. The Sobolev and RKHS assumptions are standard in learning theory and are commonly used to derive generalization guarantees. While the rates depend on dimensionality, this is inherent to the problem. Sobolev regularity helps reduce this effect and leads to near-optimal rates.
>
> *"Does this method suffer from a curse of dimensionality? Would it be feasible to present numerical experiments on systems with higher state dimensions or input parameters?"*
>
> While our experiments focus on a 2D setting, the method is theoretically designed to scale to higher-dimensional problems. The curse of dimensionality is fundamental in nonparametric learning, but our approach scales gracefully with smoothness. The convergence rates scale as $\mathcal{O}(N^{-\nu/(m+1+2\nu)})$, where $\nu$ denotes smoothness and $m$ the input dimension. Thus, the curse of dimensionality can be mitigated under sufficient regularity.
>
> The main computational cost arises from Gram matrix inversions, which can be reduced using standard approximations such as Nyström or random feature methods, without sacrificing theoretical guarantees. Appendix B provides full complexity analysis. In particular, the computational cost of our safe method is of the same order of magnitude as that of the (unsafe) standard batch kernel ridge regression.
>
> Extending experiments to higher-dimensional systems is a promising and feasible direction for future work.
>
> *"In assumptions A1 and A2, is the requirement for all theta, or there exists a theta? The former is very restrictive but appears at odds with the written text."*
>
> We acknowledge that Assumptions (A1) and (A2) may appear strong, but they are minimal in what they require: only one known safe and one known reset point are sufficient to initiate safe exploration. The sets $S_0$ and $R_0$ can be as small as singletons. Without at least one such point, safe learning cannot begin. In many systems, states of the form $S_0 = \{(0, \theta) : \theta \in D\}$ and $R_0 = \{(0, \theta) : \theta \in D\}$ are valid and realistic, as systems are often initialized in known, safe and resettable conditions.
>
> We express the assumptions in general (set-based) form to support more practical and informative choices: our approach leverages non-singleton safe and reset sets to accelerate learning while preserving formal safety guarantees.
>
> *"Can you expand more on how smoothness of dynamics relates to drift and diffusion terms?"*
>
> While we do not study this explicitly, Sobolev regularity of the drift and diffusion terms is expected to imply Sobolev regularity of the resulting state densities under standard conditions. This follows from classical results in parabolic PDE theory, where under standard assumptions, solutions gain regularity relative to the coefficients, approximately two derivatives in space and one in time. We will add a comment on this in the revised version of the paper. See Bonalli, R., and Rudi, A. (2025). Non-parametric learning of stochastic differential equations with non-asymptotic fast rates of convergence. Foundations of Computational Mathematics, 1–56, for related results.
>
> *"It is hard to parse the definition of initial safe resettable set, can you explain the role of t and t'?"*
>
> In the definition of $\Gamma_0$, $t$ denotes the planned sampling time and $T$ the reset time. The condition $(\theta, t') \in S_0$ for all $t' \in [0, T]$ ensures that the system is known to be safe at every intermediate time between sampling and reset. Thus, $t$ indicates where the observation is collected, and the condition on $t'$ ensures safety over the whole trajectory. This guarantees that a sample at $(\theta, t)$ lies within a trajectory that is entirely safe and ends in a known resettable state.
>
> As mentioned earlier, $\Gamma_0$ can be as simple as a singleton, e.g., $\Gamma_0 = {(\theta, 0, 0) \mid \theta \in D}$, corresponding to a trajectory of length zero-trivially satisfying the requirement. The general formulation allows the use of larger sets when more prior knowledge is available, enabling faster exploration.
>
> *"At the beginning of section 4.2, should it be index i instead of N? Or can you explain the role of these two indices?"*
>
> In this section, $N$ refers to the iteration index, i.e., the control $(\theta_N, t_N)$ being evaluated. The superscript in $w_i^N$ indicates that the $i$-th sample trajectory is drawn at iteration $N$ using control $u_{\theta_N}$. So $i$ indexes the sample trajectories for a fixed control, and $N$ identifies which control they correspond to. We agree this could be clarified in the notation and will improve the explanation in the revision.
>
> *"Can you further explain the assumption on N in theorem 5.1? It seems unintuitive to upper bound N by assumption. But maybe its actually better understood as a bound on Q, the number of samples per trajectory?"*
>
> We agree: the condition is better understood as a lower bound on $Q$ as a function of $N$. It ensures that the number of sample trajectories per control is large enough to guarantee accuracy. The logic is: fix a target accuracy $\eta$, then determine the required \$N\$, and then determine the required \$Q = Q(N)\$ to achieve it. We will clarify this in the revised version.
> More precisely, the assumption requires: $Q \gtrsim N^{(2\nu + n)/(2\nu - n)},$ up to log factors, where $n$ is the state dimension and $\nu$ is the regularity. This reflects the classical trade-off in kernel estimation and helps quantify how regularity mitigates the curse of dimensionality.
>
> *Can you compare the size of the learned safe region to that of the known dynamics? If not theoretically, then at least empirically?
> Why is the definition of safety meaningful? Shouldn't we care about absolute safety integrated over time, rather than just at each instant probabilistically?"*
>
> While a theoretical comparison is not provided, Figures 4, 5, and 6 show that the learned safe set expands empirically toward the reachable true region. Looser safety thresholds allow the algorithm to recover a larger portion of the reachable set-i.e., the region accessible without crossing unsafe states or requiring discontinuous jumps.
>
> Regarding the safety definition, we use the quantity
>
> $$
> s^\infty(\theta, T) = \inf_{t \in [0, T]} \mathbb{P}(g(X(t)) \geq 0),
> $$
>
> which ensures high-probability safety at each individual time point. While enforcing trajectory-level safety (i.e., $\mathbb{P}(g(X(t)) \geq 0 \ \forall t \in [0, T])$) is indeed a desirable goal, it remains fundamentally harder to verify under unknown dynamics. Our pointwise formulation provides a tractable and meaningful notion of safety, especially in the absence of model knowledge.
>
> To the best of our knowledge, no existing methods for exploration in unknown stochastic systems offer stronger or comparable theoretical guarantees. That said, we would be very interested in identifying and comparing to such approaches and would be glad to include any relevant references.
>
> *"Figures too small; label plots directly."*
>
> We will revise the figures to improve clarity and annotate plots directly with threshold values.

---

> > ### Comment · Reviewer_1MU6 · 2025-08-04
> >
> > Thanks for the reply, these clarifications are helpful and address several of my concerns. I have revised my score upwards.

---

### Official Review · Reviewer_gbjw · 2025-07-02

**Clarity:** 4
**Significance:** 2
**Originality:** 2
**Rating:** 4
**Confidence:** 4

**Summary:**

The paper tackles safe learning in controlled stochastic systems that begins with an initial safety-certified state–control set and models the unknown stochastic disturbance in an otherwise deterministic system under smoothness assumptions.
To do this, they make the following assumptions:
- there is an initial known safe set of controls
- there is an initial "reset set" from which the system can be reset to its initial state distribution. For example, a local region around the launchpad for a UAV from which I suppose someone could pick up the UAV and reset it.
- smoothness of the dynamics wrt time, state, and control

From these assumptions, they provide a method for updating a) the dynamics model, b) the safety model (e.g. safe set), and c) a reset model to estimate the probability of returning to the initial state distribution for repeated exploration.  This is all done iteratively based on discrete-time trajectory observations paired with kernel-based confidence bounds. Controls are selected such that the trajectories will reduce uncertainty in the dynamics while, with high probability, maintain safety while ending in the reset region.  This seems to be largely building on Felix Berkenkamp's work on safe exploration, with the addition of stochasticity.

**Questions:**

1. Could the authors comment on how sensitive exploration is to mis-specification of the sets from Assumptions?
2. Can this algorithm be applied to higher-dimensional and longer-horizon problems?

**Ethical Concerns:**

["NO or VERY MINOR ethics concerns only"]

**Final Justification:**

1. This paper tackles an important problem (data-efficient, safety-critical learning) relevant to different applications.
2. The paper is well-written with a clear presentation.
3. The assumptions are explicitly articulated and immediately followed by a discussion.
4. The theorem certifies that every trajectory sampled during training satisfies safety and reset constraints, while also obtaining convergence rates.

**Limitations:**

No limitations section, but assumptions are made clear.

**Quality:**

3

**Strengths And Weaknesses:**

Strengths:
1. This paper tackles an important problem (data-efficient, safety-critical learning) relevant to different applications.
2. The paper is well-written with a clear presentation.
3. The assumptions are explicitly articulated and immediately followed by a discussion.
4. The theorem certifies that every trajectory sampled during training satisfies safety and reset constraints, while also obtaining convergence rates.

Weaknesses:
1. Evaluation is restricted to a single low-dimensional synthetic environment, drawing questions about the practicality of the approach. The paper does not test higher-dimensional states or longer horizons, nor discuss scalability.
2. The algorithm requires both an initial set of controls that are guaranteed safe and a separate set that reliably reset the system. These assumptions (especially the latter) may be quite limiting in certain contexts
3. Ablation study of kernel choice or safety/reset thresholds should be included.

---

> ### Author Rebuttal · Authors · 2025-07-30
>
> We appreciate your constructive feedback. We address your points below.
>
> **(General comment).** This work takes a first step toward provably safe learning under unknown stochastic dynamics, a setting largely unexplored compared to prior work assuming known dynamics. Our focus is theoretical and methodological; code will be released for reproducibility. Experiments on more complex problems are part of ongoing work. The Sobolev and RKHS assumptions are standard in learning theory and are commonly used to derive generalization guarantees. While the rates depend on dimensionality, this is inherent to the problem. Sobolev regularity helps reduce this effect and leads to near-optimal rates.
>
> *"Evaluation is restricted to low-dimensional synthetic environment..."*
>
> We agree. Our goal is to provide a statistical analysis of the interaction between learning and safety. Theoretical guarantees in our framework explicitly scale with dimension: the convergence rates for the three estimators are of order $\mathcal{O}(N^{-\nu/(m+1+2\nu)})$, provided the number of samples per trajectory satisfies $Q \geq \mathcal{O}(N^{(2\nu + n)/(2\nu - n)})$ (up to logarithmic factors). This implies that, under sufficiently smooth dynamics (i.e., large $\nu$), the curse of dimensionality can be mitigated.
>
> Details on computational complexity are provided in Appendix B. Briefly, the per-iteration cost is $\mathcal{O}(i^3 + 3Qi)$ (versus $\mathcal{O}(i^3 + Qi)$ for standard regression), resulting in total complexity $\mathcal{O}(N^4 + QN^2)$. Under sufficient Sobolev regularity, $Q < N^2$ is enough, making evaluation cost negligible and overall complexity similar to batch kernel ridge regression (unsafe).
>
> *"Initial safe/reset sets may be limiting."*
>
> We acknowledge that Assumptions (A1) and (A2) may appear strong, but they are minimal in what they require: only one known safe and one known reset point are sufficient to initiate safe exploration. The sets $S_0$ and $R_0$ can be as small as singletons. Without at least one such point, safe learning cannot begin. In many systems, states of the form $S_0 = \{(0, \theta) : \theta \in D\}$ and $R_0 = \{(0, \theta) : \theta \in D\}$ are valid and realistic, as systems are often initialized in known, safe and resettable conditions.
>
> We express the assumptions in general (set-based) form to support more practical and informative choices: our approach leverages non-singleton safe and reset sets to accelerate learning while preserving formal safety guarantees.
>
> *"Ablation study on kernel and thresholds."*
>
> Thank you for the suggestion. While we did not include an ablation study in this version, we agree it would offer valuable insights. Our framework supports a broad class of kernels, and the safety/reset thresholds directly govern the trade-off between conservativeness and exploration. We plan to investigate these aspects more systematically in future work.
>
> *"Sensitivity to mis-specification of the sets in assumptions?"*
>
> This is an important point. The algorithm requires at least one correctly specified safe and resettable point to initiate exploration. If this point is misspecified (e.g., not actually safe or resettable), the algorithm may halt immediately, as it cannot proceed without safety guarantees. In this sense, the method is sensitive to gross mis-specification of the initial sets. However, this is standard in safe learning and is typically satisfied in practice by choosing conservative initial conditions, e.g., using $S_0 = \{(0, \theta) : \theta \in D\}$ and $R_0 = \{(0, \theta) : \theta \in D\}$, which are valid in many settings.
>
> *"Can this method be applied to higher-dimensional and longer-horizon problems?"*
>
> Yes. While our experiments focus on a 2D setting, the method's statistical and computational performance is theoretically characterized in arbitrary dimensions (see above). In particular, convergence rates depend on both the smoothness of the target functions and the dimensionality, allowing scalability under sufficient regularity. The main computational cost lies in kernel regression, which can be mitigated using standard techniques such as Nyström approximations or random features, without sacrificing theoretical guarantees (See, e.g. Rudi, A., Camoriano, R., and Rosasco, L. (2015). Less is more: Nyström computational regularization. Advances in neural information processing systems, 28).

---

> > ### Comment · Reviewer_gbjw · 2025-08-04
> >
> > Thanks for the detailed response! I maintain my positive score.

---

### Official Review · Reviewer_o5R2 · 2025-07-03

**Clarity:** 3
**Significance:** 3
**Originality:** 3
**Rating:** 4
**Confidence:** 2

**Summary:**

To enable safe exploration and learning in control systems subject to noise disturbances, this paper proposes a novel safe learning framework that incorporates three key components: a dynamics model, a safety model, and a reset model. The resulting safe region is theoretically guaranteed by a provable bound.

**Questions:**

1.It would be helpful if the authors could include a practical case study or provide a clearer explanation of the paper's contributions-particularly how the proposed method differs from existing safe exploration approaches.

2.Does the computation of the safety model introduce significant overhead during data collection? A discussion on the computational cost and efficiency would be beneficial.

**Ethical Concerns:**

["NO or VERY MINOR ethics concerns only"]

**Final Justification:**

1.This paper is well-written and proposes a novel safe exploration framework to address the challenges posed by the dynamic nature of controlled continuous-time stochastic systems.

2.The theoretical analysis and derivations effectively support the authors' claims.

3.The numerical experiments are solid, and the visualizations clearly illustrate the process of safe region coverage.

Since the authors effectively address my concerns, i will maintain my positive evaluation.

**Limitations:**

Yes, the author discuss limitations, but it will be helpful to include the practical discussion of this safe exploration method in real-world method.

**Quality:**

3

**Strengths And Weaknesses:**

**Strengths**：

1.This paper is well-written and proposes a novel safe exploration framework to address the challenges posed by the dynamic nature of controlled continuous-time stochastic systems.

2.The theoretical analysis and derivations effectively support the authors' claims.

3.The numerical experiments are solid, and the visualizations clearly illustrate the process of safe region coverage.

**Weaknesses**：

1.Some notations and writing could be improved for clarity. For example, w_i in Equation (5) and subsequent equations appears to represent noise but is not explicitly defined in the paper. Additionally, M_k in Equations (5) and (6) may correspond to T_{\text{max}} as defined earlier (Line 83), but this should be clarified.

2.The paper would benefit from the inclusion of an algorithm table or pseudocode to enhance clarity and reproducibility. Overall, the presentation could be improved.

3.While the experimental results demonstrate the effectiveness of the proposed method, it would be valuable to include comparisons with existing safe exploration approaches as baseline models to better prove the performance and contribution.

---

> ### Author Rebuttal · Authors · 2025-07-30
>
> Thank you for your detailed and practical comments. We address your points below.
>
> **(General comment).** This work takes a first step toward provably safe learning under unknown stochastic dynamics, a setting largely unexplored compared to prior work assuming known dynamics. Our focus is theoretical and methodological; code will be released for reproducibility. Experiments on more complex problems are part of ongoing work. The Sobolev and RKHS assumptions are standard in learning theory and are commonly used to derive generalization guarantees. While the rates depend on dimensionality, this is inherent to the problem. Sobolev regularity helps reduce this effect and leads to near-optimal rates.
>
> *"Some notations and writing could be improved..."*
>
> Thank you. We agree and will clarify that $w_i$ indexes the Brownian motion samples and that $M_k$ denotes the number of time steps in trajectory $k$. Allowing $T_k \leq T_{\max}$ enables learning controllers that are safe over shorter time horizons, improving flexibility.
>
> *"The paper would benefit from the inclusion of an algorithm table..."*
>
> We will include an algorithm table in the revised version. A well-documented Python library has already been developed and will be released as open source. Appendix B provides a detailed breakdown of each algorithmic step and its computational complexity.
>
> *"Include comparisons with safe exploration baselines."*
>
> We agree this would be valuable. Nevertheless, most existing approaches assume known dynamics and typically lack formal safety guarantees during exploration. We plan to include such comparisons in future work and welcome any relevant references we may have missed.
>
> *"Include a practical case study."*
>
> Thank you for the suggestion. We agree that practical case studies would help illustrate the relevance of our approach. Our work addresses a challenging and, to our knowledge, previously unaddressed setting: learning general unknown dynamics under safety constraints with formal guarantees. While our focus here is on establishing these theoretical foundations, the method is directly applicable to robotics domains where safety and uncertainty are central. For example, flying humanoid robots like iRonCub3 involve highly nonlinear, safety-critical dynamics, and tasks such as cloth folding require learning deformable dynamics safely (e.g., Amadio, F., Delgado-Guerrero, J. A., Colomé, A., and Torras, C. (2023). Controlled Gaussian process dynamical models with application to robotic cloth manipulation. International Journal of Dynamics and Control, 11(6), 3209–3219). We view these as promising directions for future applications.
>
> *"Does the computation of the safety model introduce overhead?"*
>
> The overhead is moderate. The per-iteration cost is $\mathcal{O}(i^3 + 3Qi)$ (compared to $\mathcal{O}(i^3 + Qi)$ for standard regression), leading to total complexity $\mathcal{O}(N^4 + QM^2)$. Under sufficient Sobolev regularity, $Q < N^2$ is enough, making the evaluation cost negligible and the overall complexity comparable to batch kernel ridge regression (unsafe).

---

> > ### Comment · Reviewer_o5R2 · 2025-08-04
> >
> > Thank you for your response, which effectively addresses my concerns. I appreciate your efforts and will maintain my positive evaluation.

---

### Official Review · Reviewer_oPjs · 2025-07-04

**Clarity:** 3
**Significance:** 2
**Originality:** 3
**Rating:** 3
**Confidence:** 4

**Summary:**

This paper proposes a method for learning the dynamics of controlled stochastic systems from trajectory data in the presense of safety constraints, and provides safety guarantees during training and deployment. The method is based on expanding a safe control set using kernel-based confidence bounds. Also, adaptive learning rates that improve with the smoothness are derived. The method is validated on a 2D second order dynamical system.

**Questions:**

If I understand it correctly, the method requires the upper bounds ||s||_G, ||r||_G, which are not obviously available (as also bounds of RKHS norms).
To increase the evaluation, it would be good to see how can these bounds be non conservatively estimated for real systems or system models? Does that impact the practical use of the method?  How sensitive are the safety guarantees to misspecification of  ||s||_G, ||r||G, and ||p||? Can you demonstrate similar performance with large underestimation of the bounds?
How are the hyparparameters β^s, β^r, λ, γ, R tuned?
Another suggestion for more convincing demonstration is to increase the complexity and dimensionality of the proposed numerical example and show how the method works in a scenario closer to reality.
Also, it would help to understand how the method performs with a comparison to safe learning control methods like model-based safe RL and control barrier functions, or more baseline method comparisons.

**Ethical Concerns:**

["NO or VERY MINOR ethics concerns only"]

**Final Justification:**

With the additional clarifications provided in the rebuttal, I find the authors are proposing a promising method.
However, the validation of the method remains limited. I therefore maintain the initial score on the paper.

**Limitations:**

- the method relies on known bounds, and it is not specified how to estimate them or how sensitive the guarantees are to misspecification
- the numerical simulation example is limited

**Paper Formatting Concerns:**

no concerns

**Quality:**

3

**Strengths And Weaknesses:**

Strengths
There is a solid theoretical analysis which provides safety guarantees and adaptive convergence rates under Sobolev regularity assumptions.
The paper is generally well-written with clear problem motivation and clear description of the approach.
The addressed problem in control of safety-critical systems is important with applications in robotics, autonomous systems, and other domains.
The paper extends safe UCB methods to continuous-time stochastic differential equations and provides joint (probabilistic) guarantees for dynamics estimation and safety.

Weaknesses

The method relies on assumptions as Sobolev regularity, RKHS membership, eventually known safe/reset sets, which might be limiting.
The validation of the method is very limited (a 2D toy example) and does not show or address the scalability of the method to more complex systems or higher dimensionality, especially in terms of computational complexity.
The comparison with other safe learning approaches is limited. It is unclear what is the impact of the reset in more realistic applications.

---

> ### Author Rebuttal · Authors · 2025-07-30
>
> Thank you for your thoughtful review. Below are our responses to the points you raised.
>
> **(General comment).** This work takes a first step toward provably safe learning under unknown stochastic dynamics, a setting largely unexplored compared to prior work assuming known dynamics. Our focus is theoretical and methodological; code will be released for reproducibility. Experiments on more complex problems are part of ongoing work. The Sobolev and RKHS assumptions are standard in learning theory and are commonly used to derive generalization guarantees. While the rates depend on dimensionality, this is inherent to the problem. Sobolev regularity helps reduce this effect and leads to near-optimal rates.
>
> *"The method relies on assumptions as Sobolev regularity, RKHS membership, eventually known safe/reset sets, which might be limiting."*
>
> We acknowledge that Assumptions (A1) and (A2) may appear strong, but they are minimal in what they require: only one known safe and one known reset point are sufficient to initiate safe exploration. The sets $S_0$ and $R_0$ can be as small as singletons. Without at least one such point, safe learning cannot begin. In many systems, states of the form $S_0 = \{(0, \theta) : \theta \in D\}$ and $R_0 = \{(0, \theta) : \theta \in D\}$ are valid and realistic, as systems are often initialized in known, safe and resettable conditions.
>
> We express the assumptions in general (set-based) form to support more practical and informative choices: our approach leverages non-singleton safe and reset sets to accelerate learning while preserving formal safety guarantees.
>
> *"Validation is limited to a 2D toy example, and scalability is not demonstrated."*
>
> We agree that evaluating the method on higher-dimensional systems would be valuable. In this paper, we focused on a low-dimensional setup to clearly illustrate the core theoretical guarantees. Higher-dimensional experiments are planned for future work. Nonetheless, the theoretical results scale with dimension, with convergence rates for the three estimators of order $\mathcal{O}(N^{-\nu/(m+1+2\nu)})$, provided the number of samples per trajectory satisfies $Q \geq \mathcal{O}(N^{(2\nu + n)/(2\nu - n)})$ (up to log factors). This means the curse of dimensionality can be mitigated under sufficient smoothness (i.e., large $\nu$), making the method applicable in higher-dimensional settings.
>
> The main computational cost arises from Gram matrix inversion, which can be reduced using standard approximation techniques such as Nyström methods, without sacrificing theoretical guarantees (see Rudi et al., 2015). Appendix B provides algorithmic details and complexity analysis. In short, the per-iteration cost is $\mathcal{O}(i^3 + 3Qi)$ (vs. $\mathcal{O}(i^3 + Qi)$ for standard regression), leading to overall complexity $\mathcal{O}(N^4 + QN^2)$. Under sufficient regularity, $Q < N^2$ is enough, making evaluation costs negligible and overall complexity comparable to batch kernel ridge regression (unsafe).
>
> *"Comparison with other safe learning approaches is limited. Impact of reset unclear."*
>
> We agree and thank you for raising this. Our setting differs from most safe RL approaches, which typically assume known dynamics or do not offer formal safety guarantees. In contrast, our method ensures certified safety during exploration under fully unknown stochastic dynamics.
>
> The reset mechanism is optional but beneficial: it prevents the learner from restarting exploration far from the initial safe region, improving sample efficiency in practice. Resets are also common in prior safe learning works, e.g., "Turchetta, M., Berkenkamp, F., and Krause, A. (2016), which rely on resets to known safe states to enable safe expansion. Safe exploration in finite Markov decision processes with Gaussian processes. Advances in neural information processing systems, 29.", which rely on resets to known safe states to enable safe expansion.
>
> Importantly, the impact of resets is reflected in our theoretical analysis. Our convergence rates depend on the Sobolev regularity of both the safety function and the reset mechanism. Thus, even if the safety constraint is highly regular, the reset function can become the bottleneck if it has lower regularity. In that sense, the reset constraint may limit the overall convergence rate.
>
> *"The method requires upper bounds on $\|s\|$, $\|r\|$, which are not always available. How sensitive are the safety guarantees to misspecification?" "How are hyperparameters $\beta^s, \beta^r, \lambda, \gamma, R$ tuned?"*
>
> We agree this is a key practical limitation.  The bounds on  $\|s\|$, $\|r\|$,  are required for our theoretical guarantees. When not known a priori, they can be conservatively overestimated, which ensures safety but may result in more cautious exploration. In some settings, such as Sobolev RKHS, these norms can be estimated from prior knowledge, for example using bounds on $|p|_{H^\nu}$.
>
> Although a quantitative analysis of how safety degrades under misspecified bounds is still missing, we believe this requirement is a reasonable trade-off for enabling provable safety under fully unknown dynamics.
>
> In our experiments, hyperparameters ($\beta^s, \beta^r, \lambda, \gamma, R$) were selected heuristically based on smoothness assumptions to reduce computational overhead (see Appendix B.3). Developing more adaptive tuning strategies to estimate these quantities more accurately is a promising direction for future work. Following Shalev-Shwartz (2012), one could explore approaches based on the doubling trick over a finite decreasing sequence of candidate values, which may help preserve theoretical guarantees without requiring prior knowledge of the quantities $\|s\|$ and $\|r\|$.
>
> Shalev-Shwartz, S. (2012). Online learning and online convex optimization. Foundations and Trends in Machine Learning, 4(2), 107–194.
>
> More broadly, this reflects a central challenge in safe learning: guaranteeing safety when both the system and its regularity are unknown. Our framework allows for variable smoothness through flexible kernel assumptions, but understanding safety under uncertainty in regularity remains an open and important problem.
>
> *"Suggestion: increase complexity and dimensionality of the experiments; compare to safe RL baselines."*
>
> We agree that evaluating on more complex, higher-dimensional environments is an important next step. This work aims to establish a principled foundation for safe learning under unknown dynamics, and scaling to more realistic domains (e.g., robotics) is part of our ongoing research.
>
> Regarding comparisons to safe RL and CBF methods: we acknowledge their relevance and value. However, to our knowledge, most existing approaches either rely on known or structured system dynamics or do not offer provable safety guarantees in the fully unknown setting we consider. We would be grateful for any specific references we may have overlooked and will gladly include relevant baselines in future versions or extended work.
>
> Several notable examples that reflect growing interest in safety-aware learning, while operating under model knowledge, include:
>
> Prajapat, M., Köhler, J., Turchetta, M., Krause, A., and Zeilinger, M. N. (2025). Safe guaranteed exploration for non-linear systems. IEEE Transactions on Automatic Control.
>
> Berkenkamp, F., Turchetta, M., Schoellig, A., and Krause, A. (2017). Safe model-based reinforcement learning with stability guarantees. Advances in neural information processing systems, 30.
>
> Turchetta, M., Berkenkamp, F., and Krause, A. (2019). Safe exploration for interactive machine learning. Advances in Neural Information Processing Systems, 32.
>
> Baumann, D., Marco, A., Turchetta, M., and Trimpe, S. (2021, May). Gosafe: Globally optimal safe robot learning. In 2021 IEEE International Conference on Robotics and Automation (ICRA) (pp. 4452-4458). IEEE.
>
> Sukhija, B., Turchetta, M., Lindner, D., Krause, A., Trimpe, S., and Baumann, D. (2023). Gosafeopt: Scalable safe exploration for global optimization of dynamical systems. Artificial Intelligence, 320, 103922.

---

> > ### Comment · Reviewer_oPjs · 2025-08-04
> >
> > Thank you for the provided clarifications.
> > On safe learning, the following work addresses leanring unknown dynamics under unknown safety constraints: Li, J., Zagorowska, M., De Pasquale, G., Rupenyan, A. and Lygeros, J., 2024. Safe time-varying optimization based on gaussian processes with spatio-temporal kernel. Advances in Neural Information Processing Systems, 37, pp.95326-95355., where the focus is on optimization, but it also applies to leanring. The work provides safety guarantees and also demonstrates the loss of optimality in the tradeoff driven by safe exploration.
> > Including a similar comparison and demonstration, even with a small example, might benefit your proposed paper by demonstrating the strength of the method.

---

> > > ### Author Response · Authors · 2025-08-04
> > >
> > > We thank the reviewer for the thoughtful feedback and for pointing out TVSafeOpt (Li et al.) as a valuable related reference. We fully agree with the reviewer that clarifying connections to existing methods helps better position our contribution.
> > >
> > > Indeed, both our work and TVSafeOpt rely on kernel-based confidence bounds. However, the two approaches differ significantly in their assumptions and objectives. Specifically, TVSafeOpt addresses deterministic dynamics, with direct (though noisy) evaluations of reward and safety, assuming known Lipschitz constants and providing guarantees primarily in stationary settings. In contrast, we consider stochastic continuous-time systems described by SDEs, where uncertainty directly affects the system's dynamics rather than only the observations. Additionally, our goal is to safely learn the underlying model itself without direct evaluations of reward or safety functions, relying instead on noisy observations of system states. We provide novel theoretical guarantees on safety and derive learning rates for system estimation, adaptively reflecting the Sobolev regularity of the dynamics. Moreover, our kernel is defined naturally over continuous time $[0, T]$, which we find particularly suitable for modeling these continuous-time systems, unlike the discrete indexing used in TVSafeOpt. While our method does not directly target reward optimization, our experiments illustrate an important trade-off between safe exploration and learning efficiency.
> > >
> > > We also appreciate the reviewer's suggestion to include additional illustrations. While our method was intentionally designed to be simple to clearly highlight its theoretical properties, we fully agree that further evaluations, particularly involving real-world scenarios, higher-dimensional dynamics, or ablation studies, would significantly strengthen its practical relevance. We view these suggestions as insightful and valuable directions for future research.
> > >
> > > We hope this clarification helps convey the scope and novelty of our contribution. We thank the reviewer again for engaging constructively with our work.

---

### Comment · Area_Chair_aBvn · 2025-08-08

Dear reviewers,

Thank you for engaging with the authors in a timely manner. Overall, I think we have had healthy author-reviewer discussions on this paper.

Could I please ask you to see if there is any input the authors can provide at this stage that would aid your decision making? Otherwise, I look forward to discussing this with you in the coming days.

---

### Note · Authors · 2025-08-11

We thank the reviewers for their feedback. This work provides the first provably safe learning framework for fully unknown stochastic dynamics with minimal assumptions and strong theoretical guarantees.

**Novelty and Distinction.** We present a principled, scalable, and provably safe approach that fills a key gap in safe learning. We obtained *near-optimal convergence rates* under standard Sobolev and RKHS assumptions. Only one known safe and one known reset state are required; these can be singletons and are realistic in many systems. Unlike most safe RL and CBF methods that assume known dynamics or lack formal guarantees, our approach ensures safety throughout exploration without model knowledge. The reset mechanism, common in prior safe learning works, is optional but beneficial, and its impact is explicitly quantified in our theory. Results scale with dimension under sufficient smoothness, mitigating the curse of dimensionality. Computational cost can be reduced via Nyström methods without loss of guarantees.

---

### Decision · Program_Chairs · 2025-09-17

**Decision:**

Accept (poster)

**Comment:**

This paper considered the task of learning an unknown controlled SDE subject to a safety constraint involving the state applied pointwise in time. Here, since the system is unknown, in addition to learning the system via discrete-time samples, the safety guarantee itself ultimately depends future evolution of states, a mapping that has to be learned. Under Sobolev continuity assumptions, the main result here is an algorithm that guarantees safety with high probability during training and deployment, while ensuring that the parameters themselves are estimated to arbitrary prespecified accuracies.

This paper has 3 positive reviews (including a clear accept) and a borderline reject. On the negative side, two of the reviewers point out that the synthetic evaluations are limited. However, all reviewers appreciated that the theoretical analysis puts together convergence/sample complexity, performance and safety rigorously in very concrete terms.

Given this, it seems like the upsides outweigh the downsides here. We are happy to recommend this paper for acceptance,